# Genetic interaction profiles of regulatory kinases differ between environmental conditions and cellular states

Siyu Sun[1,2], Anastasia Baryshnikova[3], Nathan Brandt[1,2] & David Gresham[1,2,**] (iD)

## Abstract

Cell growth and quiescence in eukaryotic cells is controlled by an evolutionarily conserved network of signaling pathways. Signal transduction networks operate to modulate a wide range of cellular processes and physiological properties when cells exit proliferative growth and initiate a quiescent state. How signaling networks function to respond to diverse signals that result in cell cycle exit and establishment of a quiescent state is poorly understood. Here, we studied the function of signaling pathways in quiescent cells using global genetic interaction mapping in the model eukaryotic cell, *Saccharomyces cerevisiae* (budding yeast). We performed pooled analysis of genotypes using molecular barcode sequencing (Bar-seq) to test the role of ~4,000 gene deletion mutants and ~12,000 pairwise interactions between all non-essential genes and the protein kinase genes *TOR1*, *RIM15*, and *PHO85* in three different nutrient-restricted conditions in both proliferative and quiescent cells. We detect up to 10-fold more genetic interactions in quiescent cells than proliferative cells. We find that both individual gene effects and genetic interaction profiles vary depending on the specific pro-quiescence signal. The master regulator of quiescence, *RIM15*, shows distinct genetic interaction profiles in response to different starvation signals. However, vacuole-related functions show consistent genetic interactions with *RIM15* in response to different starvation signals, suggesting that *RIM15* integrates diverse signals to maintain protein homeostasis in quiescent cells. Our study expands genome-wide genetic interaction profiling to additional conditions, and phenotypes, and highlights the conditional dependence of epistasis.

**Keywords** chronological aging; genetic interaction; nutrient starvation; quiescence; signaling kinase
**Subject Category** Signal Transduction
**Mol Syst Biol. (2020) 16: e9167**

## Introduction

Most cells spend the majority of their lifetime in a quiescent state defined as the temporary and reversible absence of proliferation (Lemons *et al*, 2010; O'Farrell, 2011; Valcourt *et al*, 2012). Quiescence requires exit from the mitotic cell division cycle and initiation of a distinct G0 cell cycle phase, during which cells remain viable and maintain the capacity to reinitiate the cell cycle and proliferative growth (Valcourt *et al*, 2012). In multicellular organisms, development, tissue renewal, and long-term survival are dependent upon the persistence of stem cells that are quiescent, but retain the ability to reenter the cell cycle to self-renew, or to produce progeny that can differentiate and repopulate the tissue (Miles & Breeden, 2017). Exit from quiescence, and initiation of aberrant proliferation, is a hallmark of cancer (Hanahan & Weinberg, 2011; Miles & Breeden, 2017). Conversely, many cancer-related deaths are the result of quiescent tumor cells that are resistant to therapeutics and underlie tumor recurrence (Borst, 2012; Yano *et al*, 2017). Thus, understanding cellular quiescence and how cells regulate the transition between proliferative and quiescent states is of fundamental importance to our understanding of cellular homeostasis and disease.

Cells exit the cell cycle and enter quiescence when they are deprived of essential nutrients or growth factors (Daignan-Fornier & Sagot, 2011; Klosinska *et al*, 2011; Valcourt *et al*, 2012). Quiescence in the model eukaryotic organism, *Saccharomyces cerevisiae* (budding yeast), shares many important features with that of higher organisms, including cell cycle arrest, condensed chromosomes, reduced rRNA synthesis and protein translation, and increased resistance to stress (Valcourt *et al*, 2012; Dhawan & Laxman, 2015). Therefore, the mechanisms that regulate cell cycle arrest and the establishment, maintenance, and exit from a quiescent state, as well as the physiological processes associated with this state, are likely to be shared across eukaryotic cells.

Studies of quiescence in yeast typically examine stationary-phase cells, namely cells grown to saturation in rich, glucose-containing medium (Gray *et al*, 2004; Young *et al*, 2017). In this case, cells typically first exhaust glucose through fermentative metabolism and then, following the diauxic shift, switch to respiration using ethanol as the

---

1 Center for Genomics and Systems Biology, New York University, New York, NY, USA
2 Department of Biology, New York University, New York, NY, USA
3 Calico Life Science LLC, South San Francisco, CA, USA
*Corresponding author. Tel: +1 212 998 3879; E-mail: dgresham@nyu.edu

carbon source. Upon exhaustion of ethanol, cells enter quiescence. However, in addition to carbon starvation, yeast cells respond to a variety of nutrient starvations by exiting the cell cycle and initiating quiescence (Lillie & Pringle, 1980; Gresham *et al*, 2011; Klosinska *et al*, 2011). Starvation for essential nutrients including nitrogen, phosphorus, and sulfur result in many of the same characteristics as carbon-starved cells including arrest as unbudded cells, thickened cell walls, increased stress resistance, and an accumulation of storage carbohydrates (Lillie & Pringle, 1980; Schulze *et al*, 1996; Klosinska *et al*, 2011). Although in laboratory conditions, yeast primarily experience carbon starvation, in the wild, yeast is likely to experience a diversity of nutrient deprivations. How the cell integrates these diverse signals to mount the same physiological response, and establish cellular quiescence, remains poorly understood.

The ability of stationary-phase yeast cells to maintain viability has also been used as a model for chronological aging. Chronological lifespan (CLS) has been defined as the time a yeast cell can survive in a non-dividing, quiescent state (Fabrizio & Longo, 2003; Kaeberlein, 2010; Walter *et al*, 2014). Therefore, CLS is closely related to the proportion of quiescent cells in stationary-phase cultures because non-quiescent cells have a reduced ability to reenter the cell cycle (Allen *et al*, 2006; Walter *et al*, 2014). Cells with a shortened CLS have reduced reproductive capacity upon replenishment of nutrients (Garay *et al*, 2014). Identification of genes that mediate CLS in yeast under different nutrient restrictions is potentially informative about the regulation of aging in higher organisms.

The genotype of a yeast cell has a profound impact on the regulation of quiescence. Many studies of survival in stationary-phase cells, and their application to the study of CLS, have been conducted using auxotrophic strains. However, starvation for an engineered auxotrophic requirement is an unnatural starvation that results in a failure to effectively initiate a quiescent state and therefore leads to a rapid loss of viability (Boer *et al*, 2008; Gresham *et al*, 2011). This is likely due to the fact that yeast cells have not evolved a mechanism for sensing and responding to laboratory-engineered auxotrophic requirements. The use of undefined media and auxotrophic strains for studying CLS can be confounded by inadvertent starvation for auxotrophic requirements. Thus, the identification of mutants that suppress the rapid loss of viability upon undefined starvation in auxotrophic strains may be of limited relevance for understanding the regulation of quiescence and CLS. Previous studies of quiescence using prototrophic yeast cells, and defined starvation for nutrients that are essential for growth in wild-type cells (i.e., natural starvation), have shown that the genetic requirements for quiescence differ depending on the nutrients for which the cell is starved (Gresham *et al*, 2011; Klosinska *et al*, 2011). However, whether the genes required for proliferation in different nutritional conditions are the same set of genes that are required for programming quiescence is not known.

Multiple evolutionarily conserved nutrient sensing and signal transduction pathways, including the target of rapamycin complex I (TORC1), protein kinase A (PKA), adenosine monophosphate kinase (AMPK), and PHO pathway, have been shown to regulate quiescence. The integrator of these diverse signaling pathways is thought to be the protein kinase RIM15, a great wall kinase that is a homologue of the mammalian gene, microtubule-associated serine/threonine-like kinase (MASTL) (Castro & Lorca, 2018). RIM15 appears to be downstream of multiple signaling pathways and is required for the establishment of quiescence (Broach, 2012; de Virgilio, 2012). However, how different starvation signals are coordinately transduced via these pathways, and how RIM15 orchestrates the establishment of cellular quiescence are not known.

The relationship between different cellular processes and pathways can be investigated using a variety of methods that identify physical and functional interactions. One efficient approach to define interactions between genes and pathways is through quantitative genetic interaction mapping (Billmann *et al*, 2016, 2018; Costanzo *et al*, 2016). A genetic interaction is a relationship between two genes in which the phenotype of the double mutant diverges from that expected on the basis of the phenotype of each single mutant (Tong *et al*, 2004; Boone *et al*, 2007; Mani *et al*, 2008; Beltrao *et al*, 2010; Costanzo *et al*, 2010). Genetic interactions can be informative of the functional relationship between the encoded products. Positive genetic interactions may be indicative of genes that exist within pathways or complexes, whereas negative genetic interactions often reflect genes that function in parallel pathways or processes that converge on the same function (van Leeuwen *et al*, 2017). Extension of genetic interaction mapping to test genome-wide interactions between defined alleles results in a genetic interaction profile, comprising the set of negative and positive genetic interactions for a given gene. The systematic application of this approach has demonstrated that genes that share similar functions, or operate in the same pathway, often share similar genetic interaction profiles. As such, the similarity in quantitative genetic interaction profiles between two genes (typically quantified as a correlation coefficient) is informative about the similarity between the two genes' functions. The culmination of genome-wide genetic interaction mapping in budding yeast has been the construction of a global genetic interaction similarity network that serves as a functionally informative reference map (Costanzo *et al*, 2010, 2016). The recent completion of this comprehensive genetic interaction map leads to two related questions: (i) To what extent are genetic interactions dependent on environmental conditions? and (ii) can genome-wide genetic interaction mapping be expanded to other phenotypes? Quantitative genetic interaction mapping is increasingly being applied in other organisms, including *Drosophila melanogaster* and mammalian cells using RNAi or CRISPR/Cas9 (Fischer *et al*, 2015; Billmann *et al*, 2016, 2018; Du *et al*, 2017; Horlbeck *et al*, 2018; Norman *et al*, 2019 making these questions of broad significance.

To date, genome-wide genetic interaction mapping in yeast has primarily been performed in a single condition and assayed using a single phenotype—colony growth—in an optimal nutritional condition (Tong *et al*, 2001; Roguev *et al*, 2008; Costanzo *et al*, 2016). Some studies have extended genetic interaction mapping to different stress conditions (St Onge *et al*, 2007; Gutin *et al*, 2015; Martin *et al*, 2015; Díaz-Mejía *et al*, 2018), but not on a genome-wide scale. Therefore, the extent to which genetic interactions depend on environmental conditions and the feasibility of using additional phenotypes beyond colony growth phenotypes in genetic interaction mapping remains largely unexplored. Targeted studies of specific genotypes suggest that functional relationships between genes are environmentally dependent (St Onge *et al*, 2007; Bandyopadhyay *et al*, 2010; Díaz-Mejía *et al*, 2018; Jaffe *et al*, 2019), suggesting that a complete understanding of global genetic interaction networks requires identification of genetic interactions in multiple conditions and using multiple phenotypes.

Here, we have developed a new method for quantifying phenotypes of pooled single and double mutants in different conditions using Bar-seq. We applied this approach to quantify the genetic requirements, and identify genetic interactions, in two different cellular states and three different nutritional conditions. Our experimental design entailed quantification of both fitness during proliferative growth and survival during prolonged defined starvation for each genotype. We find that the genetic requirements for quiescence differ depending on the nutrient starvation signal. Using genome-wide genetic interaction mapping for three key regulatory kinases, we find that these genes exhibit different interaction profiles in different growth conditions and in different cellular states. Finally, we find that the master regulator of quiescence, *RIM15*, shows distinct genetic interaction profiles and regulates different functional groups in response to different starvation signals. However, vacuole-related functions show consistent genetic interactions with *RIM15* in response to different starvation signals consistent with *RIM15* controlling quiescence by integrating diverse signals to regulate protein degradation processes (Cameroni *et al*, 2004; Swinnen *et al*, 2006). *RIM15* also interacts positively with ERAD genes specifically in nitrogen starvation conditions pointing to a previously unappreciated role for this quality control pathway in quiescence. Our study points to a rich spectrum of condition-specific genetic interactions that underlie cellular fitness and survival across a diversity of conditions and introduces a generalizable framework for extending genome-wide genetic interaction mapping to diverse conditions and phenotypes.

## Results

### Quantifying mutant fitness using pooled screens in diverse conditions

Cellular quiescence in yeast can be induced through a variety of nutrient deprivations, but whether establishment of a quiescent state in response to different starvation signals requires the same genetic factors and interactions is poorly understood. Therefore, we sought to test the specificity of gene functions and genetic interactions in quiescent cells in response to three natural nutrient starvations: carbon, nitrogen, and phosphorus. The use of prototrophic yeast strains is essential for the study of quiescence as unnatural (starvation of an auxotroph for its auxotrophic requirement), or unknown starvations can confound results and their interpretation (Boer *et al*, 2008; Gresham *et al*, 2011). Therefore, we constructed haploid prototrophic double mutant libraries using a modified synthetic genetic array (SGA) mating and selection method (Fig EV1A). Briefly, double mutant libraries were constructed using genetic crosses between the ~4,800 non-essential gene deletion strains (Giaever *et al* 2002) and query strains deleted for one of three genes encoding the catalytic subunit of different regulatory protein kinases: *TOR1*, *RIM15*, and *PHO85* (Table EV1 and Materials and Methods). In addition, we constructed a single mutant library using the same method by mating the gene deletion collection with a strain deleted for *HO*, which has no fitness defects in haploids. We confirmed the genotype and ploidy of the resulting three haploid double gene deletion libraries and the single mutant (*HO*) library using selective media and flow cytometry (Fig EV1B).

Previously, genome-wide genetic interaction mapping in yeast has been performed using colony growth phenotype as a measurement of genotype fitness (Costanzo *et al*, 2010, 2016). In liquid cultures, the growth cycle of a population of microbial cells comprises a lag period, an exponential growth phase, and a subsequent period in which growth is no longer observed, known as stationary phase. Stationary phase is indicative of cell growth and cell cycle arrest due to starvation for an essential nutrient (de Virgilio, 2012). To study each genotype over the complete growth cycle in liquid cultures, we first analyzed the four libraries (Fig 1A) in three different defined nutrient-restricted media: carbon-restricted (minimal media containing 26.64 mM carbon), nitrogen-restricted (minimal media containing 0.8 mM nitrogen), and phosphorus-restricted (minimal media containing 0.04 mM phosphorus) (Table EV2 and Materials and Methods). The composition of these media ensures that, following an exponential growth phase, cells experience either carbon, nitrogen, or phosphorus starvation, respectively. In each of the three media, $1.5 \times 10^8$ cells from each of the four libraries (Fig 1A) of pooled mutants were used to inoculate cultures ($t = 0$). In nitrogen- and phosphorus-restricted media, we observed that the starvation period commenced 24 h after inoculation (Fig EV1C). Cells in carbon-restricted media underwent the diauxic shift after 24 h and reached stationary phase approximately 48 h post-inoculation (Fig EV1C). Beyond these time points, we did not observe additional cell division or population expansion consistent with defined nutrient starvation and the initiation of quiescence.

To compare the fitness of each genotype over the complete growth cycle in each condition, a 1 ml sample ($1 \times 10^6$ cells) was removed from the culture at sequential time points and the subpopulation of viable cells was expanded using 24–48 h of outgrowth in supplemented minimal media (Fig 1A and Materials and Methods). This step is required to enrich for mutants that survive proliferation and starvation and to deplete those that have undergone senescence. Sampling $1 \times 10^6$ cells from the cultures minimized the probability ($P < 0.018$) that a genotype was not measured due to sampling error (Fig EV1D and Materials and Methods). We also quantified population viability throughout this period and observed no substantial change in any of the conditions (Fig EV1E). Using an identical outgrowth step at every time point, and determining the rate of change in the relative abundance of viable mutants in the outgrown population, accounts for growth rate differences between mutants during the outgrowth (Gresham *et al*, 2011). The abundance of each mutant in the heterogenous pool was estimated by sequencing DNA barcodes that uniquely mark each genotype using Bar-seq (Smith *et al*, 2009; Robinson *et al*, 2014; Costanzo *et al*, 2016). In total, we studied ~4,000 strains that passed filtering (Material and Methods) in each of the four libraries in the three conditions with between 3 and 5 independent experiments to account for biological and technical variability (i.e., total of 39 genetic screens).

To determine the fitness of each strain during the complete growth cycle, we initially applied linear regression modeling of the relative frequency of each mutant against time ($t = 0$, 24, 48, 96, 186, and 368 h) (Fig EV1C and Dataset EV1). To test the reproducibility of our fitness assay, we first estimated fitness for each biological replicate separately and used principal component analysis (PCA) to identify and exclude poorly behaved libraries (Fig EV1F). Hierarchical clustering of the filtered libraries shows

that, for all 39 experiments, biological replicates cluster as nearest neighbors (Fig 1B). Different libraries cultured in the same medium tend to cluster together, indicating that environmental conditions are a major determinant of fitness effects (Fig 1B). However, the *PHO85* library in nitrogen-restricted media and the *RIM15* library in phosphorus-restricted media were exceptions to this general trend (Fig 1B), which indicates that genotype also plays a key role in determining fitness. In general, mutants in carbon-restricted media

show less similarity to that observed in nitrogen- and phosphorus-restricted conditions, particularly for *HO* and *PHO85* libraries (Appendix Fig S1).

To quantify fitness, and the associated uncertainty (expressed as a 95% confidence interval), for each estimate we performed model fitting for each library in each condition using all biological replicates. We identified numerous cases in which the fitness of a single mutant significantly differs between conditions. For example,

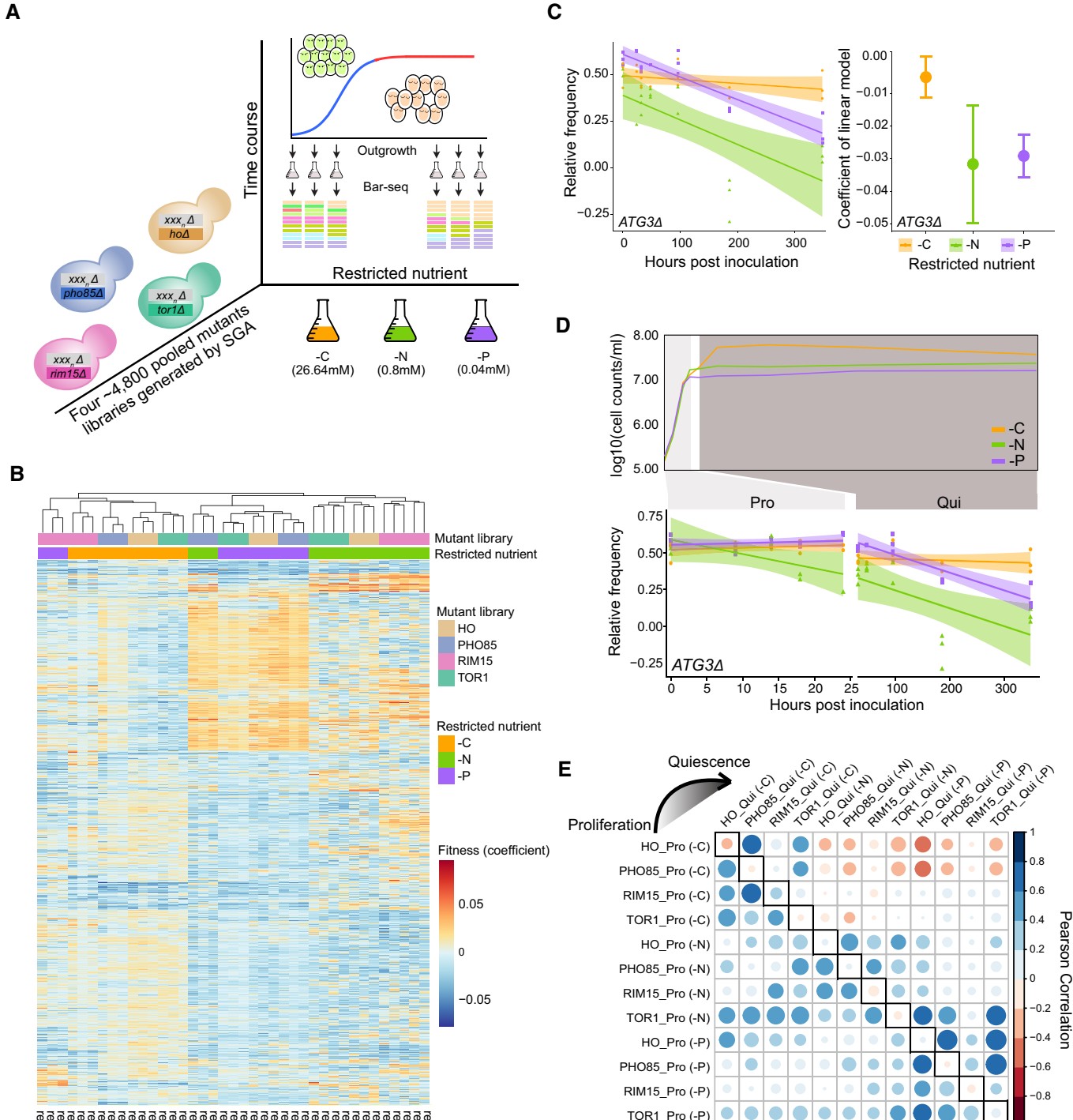

**Figure 1.**

◀

**Figure 1.  Fitness and survival rate estimation over the entire growth cycle using pooled mutant libraries and Bar-seq.**

A   Experimental design for multiplexed mutant survival assay using Bar-seq. The synthetic genetic array (SGA) method was used to construct four genome-wide double mutant prototrophic libraries (Fig EV1A). The yeast deletion collection (*xxx$_n$::natMX*) was mated with query strains deleted for one of three genes that encode regulatory kinases important in quiescence: *TOR1 (tor1Δ::kanMX)*, *RIM15 (rim15Δ::kanMX)*, and *PHO85 (pho85Δ::kanMX)*. A control library was made by mating the deletion collection to a neutral gene deletion of *HO (hoΔ::kanMX)*. To maintain library complexity, $1.5 \times 10^8$ cells from each library were used to inoculate ($t = 0$) cultures restricted for glucose (-C, 26.64 mM), ammonium sulfate (-N, 0.8 mM), and potassium phosphate (-P, 0.04 mM) in 300-ml cultures. The starvation period for -N and -P conditions commenced after 24 h and after 48 h for -C condition (Fig EV1C). At different time points, we removed a $\sim 1 \times 10^6$ cell sample from the culture and expanded the viable subpopulation using outgrowth in supplemented minimal media (Table EV2). DNA was isolated from the resulting outgrowth culture, and the library composition was analyzed using Bar-seq.

B   Hierarchical clustering of mutant fitness profiles computed for each replicate separately across the entire culturing period. White indicates that the strain has not changed in fitness compared to wild type, blue represents increased fitness, and red represents decreased fitness. Culture conditions are indicated by color (orange: carbon-restricted; green: nitrogen-restricted; and purple: phosphorus-restricted). Three kinase mutant libraries (*TOR1*, *RIM15*, and *PHO85*) and one control library (*HO*) are shown.

C   Representative gene (*ATG3*) for relative fitness estimation across the entire culturing period. The abundance of the *atg3Δ0 hoΔ0* strain was determined at multiple time points on the basis of counts of its unique DNA barcode, and fitness was determined using linear regression. Linear models (predicted value ± 95% CI) fit to the data are shown on the left, colored by condition. The coefficient (slope) of each model is shown in the dot plot on the right, with a 95% confidence interval indicated as an error bar (-C: $n = 17$, -N: $n = 9$, and -P: $n = 17$).

D   Cells exist in two distinct states depending on nutrient availability. An example of fitness determined during proliferation, and survival determined during quiescence, in the three different nutrient-restricted conditions is shown for *atg3Δ0 hoΔ0*.

E   Relationship of fitness profiles and survival profiles between mutant libraries. Heatmap of correlation coefficients between fitness profiles (lower left) and between survival profiles (upper right) for four different mutant libraries (*HO*, *TOR1*, *RIM15*, and *PHO85*) in three nutrient-restricted conditions (carbon—C, nitrogen—N, and phosphorus—P) and two cellular states (Pro and Qui). The dots on the diagonal (solid box) indicate the correlation between fitness and survival profiles for the same mutant library under the same nutrient restricted condition. Both the color and size of each dot reflect the pearson correlation.

deletion of the autophagy gene, *ATG3* (*atg3Δ0 hoΔ0*), results in reduced fitness in nitrogen- and phosphorus-restricted media, but not in carbon-restricted media (Fig 1C).

## Nutrient starvation signal is the primary determinant of mutant survival in quiescent cells

The fitness of a genotype during proliferative growth in different media may differ from the survival of the genotype in response to a specific starvation signal. To test this, we separately modeled the relative abundance of each genotype during the growth phase (i.e., from $t = 0$ to $t = 24$ h) and during the starvation period (i.e., from $t = 32$ to $t = 368$ h) for all mutant libraries using all replicates. This analysis distinguishes the effect of each gene deletion in two distinct physiological states: proliferation and quiescence. As cells do not generate progeny when starved, we refer to the phenotype during the starvation phase as "survival" and phenotype during proliferation as "fitness" (Fig 1D). To identify the primary determinant of these two phenotypes, we quantified the similarity between fitness and survival for each mutant library in each condition (C-, N-, and P-restricted conditions) (Fig 1E and Table EV3). We find a clear distinction between proliferative and quiescent cells. During proliferation, mutant libraries tend to share similar fitness profiles regardless of the nutritional condition (Fig 1E, lower left). By contrast, in quiescent cells, different mutant libraries starved for the same nutrient tend to have more similar survival profiles than the same library starved for different nutrients (Fig 1E, upper right). Consistent with the fitness estimates over the entire growth cycle, libraries starved for carbon have negative correlation with the libraries starved for nitrogen and phosphorus (Fig 1E and Appendix FigS2).

## Distinct cellular functions are required for quiescence in response to different nutrient starvation signals

Previous genome-wide genetic analyses of quiescence quantified the survival of each mutant in the absence of specific essential nutrients

but did not assess the effect of each gene deletion on cellular proliferation prior to starvation (Klosinska *et al*, 2011). To test whether the genetic requirements for proliferation in nutrient-restricted media and quiescence in response to starvation for the same nutrient are distinct, we investigated the fitness and survival of each genotype in the single mutant library (i.e., the *HO* library). We find that fitness in proliferation and survival in quiescence are poorly correlated for all three nutrient-restricted media: Pearson $r = -0.033$ in carbon-restricted condition, 0.052 in nitrogen-restricted conditions, and 0.064 in phosphorus-restricted conditions (Fig EV2A and Dataset EV2). The fitness of the single gene deletion mutants (Materials and Methods) is distributed around 0 in each of the three proliferative conditions (Fig 2A), and the majority of mutants do not show significant fitness defects compared to wild-type cells (Figs 2A and EV2B). By contrast, we find that many mutants show a survival defect in quiescent cells when starved for specific nutrients (Fig EV2B), resulting in increased variance in the distributions of survival compared to the distributions of fitness (Fig 2A). Many of the genes that are dispensable for proliferative growth in each of the three media conditions are required for quiescence. For example, deletion of genes involved in the cAMP-PKA signaling pathways, GPB1/2, RGT2, and GPR1, results in a profound survival defect in response to carbon starvation, but deletion of these genes does not lower the fitness of carbon-restricted proliferating cells and instead appear to result in a fitness increase, suggestive of a trade-off (Fig 2A, left panel). This observation is consistent with the fact that mutations in cAMP-PKA pathway have increased fitness in experimental evolution performed in carbon-limiting conditions (Venkataram *et al*, 2016). Similarly, the autophagy genes *ATG4, ATG5, ATG7*, and *ATG12* have poor survival when starved for nitrogen, but do not have a fitness defect during proliferation in nitrogen-restricted media (Fig 2A, mid-panel). In response to phosphorus starvation, genes involved in response to pH have poor survival, but those same mutants show fitness increase in phosphorus-restricted proliferating cells (Fig 2A, right panel). Thus, the genetic requirements for growth in a specific nutrient-restricted

media and quiescence in response to starvation for that nutrient are distinct.

To further investigate the functional relationship between proliferating and quiescent cells, we applied Gene Set Enrichment Analysis (GSEA) (Subramanian *et al*, 2005; Yu *et al*, 2012) to the fitness and survival profiles of the single mutant library in each nutrient-restricted condition. We find no functional overlap between different cellular states under the same nutrient-restricted condition (Fig EV2C). For example, deletion of genes involved in protein

deacetylase activity shows no significant impact on survival in quiescent cells, but results in reduced fitness during proliferation in nitrogen-restricted conditions (Fig EV2C).

Genes may be required specifically for proliferation, specifically for quiescence, or necessary for both. To identify gene functions that have a critical role uniquely in quiescence, we performed GSEA using gene lists ranked by the phenotypic difference between survival in quiescent conditions and fitness in proliferative conditions ($S_{Qui} - F_{Pro}$) (Materials and Methods and Dataset EV3). We

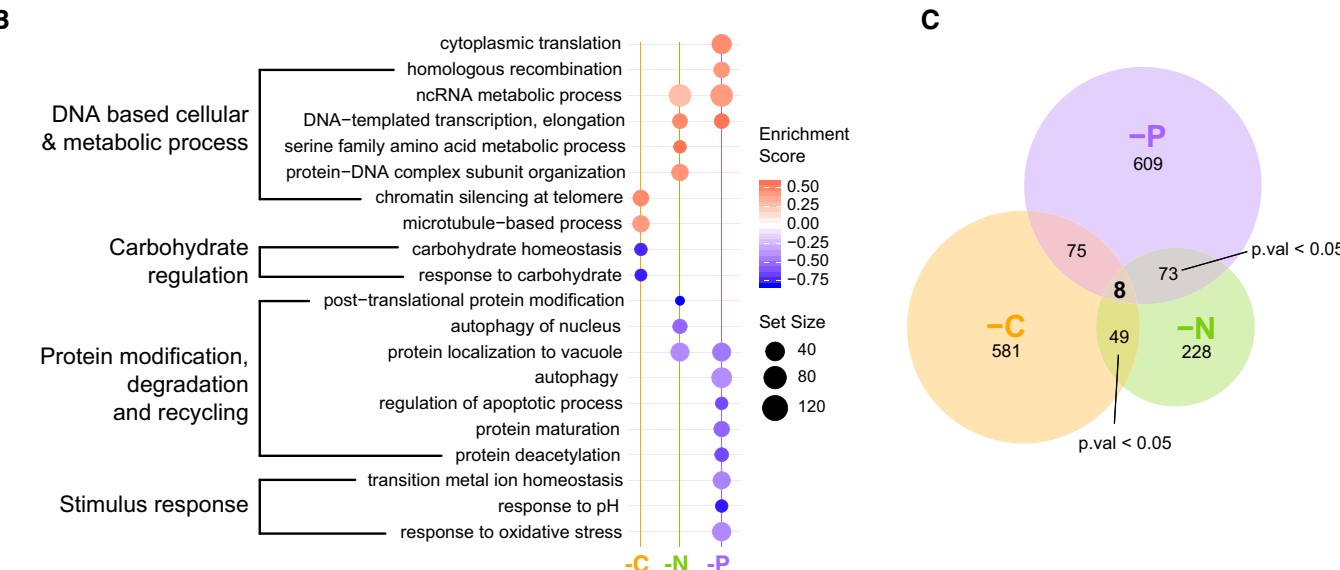

**Figure 2. Distinct functional requirements for quiescence in response to different starvation signals.**

A  Violin plots of the distribution of fitness and survival for all mutants during proliferation and quiescence in response to different nutrient restrictions. The indicated genes are examples of genes that are dispensable for proliferative growth (increased fitness or no significant fitness defects) in each of the three conditions but required for quiescence (decreased survival).

B  Enriched GO terms identified using Gene Set Enrichment Analysis (GSEA). GSEA was applied to a ranked gene list based on the difference in survival during starvation and fitness during proliferation ($S_{Qui} - F_{Pro}$) estimated using ANCOVA. The false discovery rate (FDR) was set at 0.05. Positive enrichment scores (red) indicate functions that have increased survival when starved ($S_{Qui} - F_{Pro} > 0$). Negative enrichment scores (blue) indicate functions that when impaired result in decreased survival ($S_{Qui} - F_{Pro} < 0$) during nutrient starvation. Set size indicates the gene number in each enriched term.

C  Genes that are required for quiescence but dispensable for proliferation. We found 8 genes that are commonly required for survival of all three nutrient starvations (Fig EV2D); however, the overlap between conditions is not significant (Fisher's exact test, $P > 0.05$).

identified significantly enriched GO terms ($P_{adj} < 0.05$) and found that functions involved in responding to the specific starvation signal are required for survival. For example, trehalose accumulation provides a reserve of fermentable sugar to reinitiate the cell cycle and provides protection against stress in quiescence (Gray *et al*, 2004; Shi *et al*, 2010; Klosinska *et al*, 2011). Therefore, we expect to see mutants defective in trehalose storage should fail to survive when starved for carbon. Indeed, this is the case, but the impairment of this function does not impact survival when starved for nitrogen or phosphorus (Fig 2B and Table EV4). Autophagy has previously been found to affect survival during phosphorus starvation (Gresham *et al*, 2011), which we recapitulate in our assay (Fig 2B). Similarly, we find that genes required for survival of nitrogen starvation are uniquely enriched for selective autophagy of nucleus-related amino acid trafficking and recycling (Fig 2B) consistent with protein degradation involving autophagy playing a major role in response to nitrogen starvation (Tesnière *et al*, 2015). Some functional groups show similar requirements in response to both nitrogen and phosphorus starvations, such as autophagy and protein localization by the cytoplasm-to-vacuole targeting (CVT) pathway. By contrast, response to carbon starvation requires an entirely unique set of gene functions. Thus, the biological pathways and functions that are specifically required for cellular quiescence differ depending on the nutrient starvation signal.

**No evidence for common quiescence-specific genes**

We sought to determine whether a common set of genes are required for quiescence in all starvation conditions. We identified a comparable number of quiescent-specific (hereafter: QS) genes detected in carbon (713) and phosphorus (765) restriction media. In nitrogen-restricted media, we identified about 2 times fewer QS genes: 358 (Fig 2C and Table EV5). To define a common set of QS genes, we applied three independent filtering criteria. We identified mutants (i) that are dispensable for proliferation in all three nutrient-restricted conditions ($F_{Pro} \geq 0$, $P_{adj} < 0.05$), (ii) that show significant defects in quiescence in all three conditions ($S_{qui} < 0$, $P_{adj} < 0.05$), and (iii) for which there is a significant negative difference between fitness and survival in all three conditions ($S_{qui} - F_{Pro} < 0$, $P_{adj} < 0.05$) (Materials and Methods). Using these criteria, we found 8 genes that are commonly required for quiescence regardless of the type of nutrient starvation (Figs 2C and EV2D). However, this does not differ from what would be expected by chance (Fisher's exact test, $P > 0.05$). Thus, we find no evidence for the existence of a common set of QS genes that are required for establishing quiescence in response to carbon, nitrogen, and phosphorus starvations.

**Detection of genetic interactions using pooled assays**

We aimed to identify the set of genetic interactions between all non-essential genes and the three query kinase genes in three different nutritional conditions (carbon-, nitrogen-, and phosphorus-restricted media) and two different cellular states (proliferation and quiescence). As there have been limited studies using pooled fitness assays and time course data for quantifying genetic interactions, we considered two possible approaches to do data analysis. First, we used analysis of covariance (ANCOVA) to compute the genetic

interaction score (GIS) defined as the fitness (in proliferation) or survival (in quiescence) difference between the double mutant (*queryΔ::kanMX xxx_nΔ::natMX*) and single mutant (*hoΔ::kanMX xxx_nΔ::natMX*) (Materials and Methods). In this case, the two different genotypes (single and double mutant) are treated as independent categorical variables in the model, scaled time is the covariate, and the normalized frequency at different time points is the dependent variable.

In a second approach, the GIS was calculated using the approach employed in previous genome-wide SGA studies, which defines a null model based on a multiplicative hypothesis and defines a genetic interaction as a significant difference ($\varepsilon$) between the observed and expected double mutant fitness: $\varepsilon = f_{ab} - f_a \cdot f_b$ (Costanzo *et al*, 2010). We computed the expected fitness for each double mutant by first computing the two single mutant fitness from the single deletion library (*HO*) and then computing $\varepsilon$ by determining the difference between the expected and measured fitness of double mutants (Materials and Methods). A limitation of this approach is that both single gene deletion mutants must be well measured, whereas the ANCOVA approach does not require quantifying the query mutant in the single mutant library and therefore only requires the measurement of one single mutant phenotype.

We find that the agreement between the two approaches is high (Pearson $r > 0.9$) when applied to both fitness in proliferative cells and survival in quiescent cells. The genetic interaction profiles calculated by ANCOVA or the multiplicative model for both *TOR1* and *RIM15* (Fig EV3A and B) are highly correlated across all nutrient-restricted conditions (Dataset EV4). As the *PHO85* deletion allele was not identified in the single mutant library (possibly due to an erroneous barcode), we could not perform this comparison for *PHO85* genetic interactions. To further compare the two approaches, we applied GSEA to genetic interaction profiles calculated using each model and compared the similarity of generated GO terms using GoSemSim (Yu *et al*, 2010). The significant GO terms for a given condition identified using the different models are very similar (Table EV6), indicating that ANCOVA identifies the same genetic interactions and functional groups as the classic multiplicative model. An analysis of estimated effect sizes and standard error indicates that there are no systematic biases in applying the ANCOVA model (Appendix Fig S3). As ANCOVA has a well-developed statistical framework for error estimation and significance testing, we elected to use ANCOVA to compute GIS for all subsequent analyses. Using this approach we find that gene deletions with larger phenotypic effects (either fitness or survival) tend to have strong interactions (Appendix Fig S4) as has been previously observed (Costanzo *et al*, 2016).

**Genetic interactions are condition-dependent and common in quiescence**

To date, genome-wide genetic interaction mapping in yeast has primarily been performed in a single condition (rich media) and assayed using a single phenotypic readout—colony size. To investigate the utility of using additional phenotypes in genetic interaction mapping, we compared both fitness estimates and genetic interactions identified in our study (Dataset EV5) with the global reference set (Costanzo *et al*, 2016). As our conditions (nutrient limitation and nutrient starvation) differ substantially from those used in the

global reference set the genetic requirements are likely to be distinct. Indeed, we find no significant correlation between fitness measurements in all conditions (three proliferative and three quiescent conditions) assayed in our study and those of (Costanzo *et al*, 2016) (Appendix Fig S5), supporting the notion that fitness effects are highly conditionally dependent. Similarly, no significant correlation was detected between the genetic interaction profiles quantified in both studies (Appendix Fig S6). To test whether this reflects different sources of noise in two assays, we considered only those significant genetic interactions that were identified in both Costanzo *et al* (2016) and our study. However, we find that there is poor agreement, even for genetic interactions identified in our carbon-restricted proliferative conditions, which is the most analogous condition to rich undefined media (Appendix Fig S6).

We find that genetic interactions between genes are also frequently condition-dependent and differ as a function of both cellular state and environmental conditions. For example, in quiescent cells, the autophagy gene *ATG7* positively interacts with *TOR1* in carbon starvation, but negatively interacts with *TOR1* in phosphorus starvation (Fig 3A and B). *ATG7* interacts negatively with *PHO85* and *RIM15* in phosphorus starvation, but these interactions are not found in carbon or nitrogen starvation conditions (Fig 3A and B). This example is illustrative of the conditional dependence of genetic interactions, which we find is the case for the vast majority of genotypes (data and model fitting for all genetic interactions can be explored in the associated web application, http://shiny.bio.nyu.edu/ss6025/shiny_Genetic_Interaction/).

We find a weaker correlation between phenotypes of single and double mutants in quiescent cells compared with proliferative cells (Figs 3C and EV3C). More genetic interactions are detected in quiescent cells compared to proliferative cells regardless of the starvation signal (Fig 3D). For example, at an FDR of 5%, 55 genes (~1.4% of mutant pairs tested) show significant interactions with *TOR1* in proliferative cells growing in carbon-restricted media (Fig EV3D). The fraction of genes that significantly interact with *TOR1* is similar to the proportion of significant interactions in Costanzo *et al* (2016). By contrast, we identified 228 negative and 381 positive (15% or ~10 times more) genetic interactions with *TOR1* in carbon-starved quiescent cells (Fig EV3D). This trend is observed for all three kinases (*TOR1*, *RIM15*, and *PHO85*) in all starvation conditions (Fig EV3D). We detected both positive and negative interactions for each of the three kinases and an increase in total interactions for a given kinase as more conditions are assayed (Figs 3D and EV3E), indicating that each additional assay reveals unique genetic interactions. We did not detect a bias in the number of positive or negative interactions in either cellular state.

### Genetic interaction profiles of kinases differ between cellular states

Genes that are functionally related tend to share a common set of genetic interactions that define a genetic interaction profile (Costanzo *et al*, 2010, 2016). As the activity of regulatory kinases depends on environmental signals, the functional consequences of deleting kinases are likely to be conditionally dependent, which may result in condition-dependent genetic interaction profiles. To identify the primary determinant of genetic interaction profiles in our study, we quantified the similarity between all pairs of genetic

interaction profiles for each kinase in each condition and cellular state. Clustering of genetic interaction profiles reveals a clear distinction between proliferative and quiescent cells (Fig EV4A).

In quiescent cells, genetic interaction profiles of the different kinases cluster as a function of the starvation signal (Fig EV4A). By contrast, in proliferative conditions *TOR1*, *RIM15*, and *PHO85* genetic interaction profiles do not exclusively cluster by nutritional condition (Fig EV4A). These results indicate that genetic interaction profiles differ as a function of cellular state and that the impact of the environmental conditions on genetic interactions is variable.

To visualize the correlation between genetic interaction profiles for each kinase in each condition, we constructed correlation networks for both proliferative and quiescent cells (Fig 4 and Table EV7). These correlation networks emphasize the importance of cellular state in determining the similarity of genetic interaction profiles as the genetic interaction profile similarity network is drastically remodeled in quiescence compared to proliferation (Fig 4). For example, a negative correlation is detected between *TOR1* and *PHO85* genetic interaction profiles in proliferative cells growing in carbon-restricted condition, but their genetic interaction profiles are positively correlated in carbon-starved quiescent cells (Figs 4 and EV4B). For cells in the same physiological state, the environmental conditions can also alter the relationship between the same pair of kinases. For example, *RIM15* and *PHO85* genetic interaction profiles are highly correlated during growth in carbon-restricted media, but this similarity is greatly reduced during proliferation in phosphorus-restricted conditions (Figs 4 and EV4C). These results suggest that environmental conditions alter the regulatory relationships among signaling pathways both in quiescent and in proliferative cells.

### Genetic interaction profiles are functionally coherent

To functionally annotate genetic interaction profiles for each kinase in each condition, we used spatial analysis of functional enrichment (SAFE) (Baryshnikova, 2016). We used SAFE to map quantitative attributes onto the reference network, defined by the correlation network of genome-wide genetic interaction profiles of 3,971 essential and non-essential genes (Costanzo *et al*, 2016), and tested for functional enrichment within densely connected regions, which define domains. Each of the 17 domains within this map comprises genes that share similar genetic interaction profiles and functional annotations (Fig EV5A). Importantly, SAFE uses the entire set of genetic interactions for a given query gene (Dataset EV6), including those interactions that do not reach statistical significance, which allows identification of coherent trends that may exist despite a lack of significance associated with each individual genetic interaction. This analysis tests specifically for coherence in attributes such that strong positive and negative genetic interaction scores that are randomly distributed throughout the network will produce no enrichment, whereas weak scores that tend to cluster as either positives or negatives within domains will have significant enrichment. We superimposed genetic interaction profiles of each kinase in each of the three nutrient-restricted media and both cellular states onto the reference network using SAFE. We find that kinases that show higher similarity in genetic interaction profiles (Fig 4) also show more similar enrichment patterns using SAFE analysis (Fig 5). In general, genetic interactions in proliferative conditions tend to show increased enrichment when superimposed on this reference map

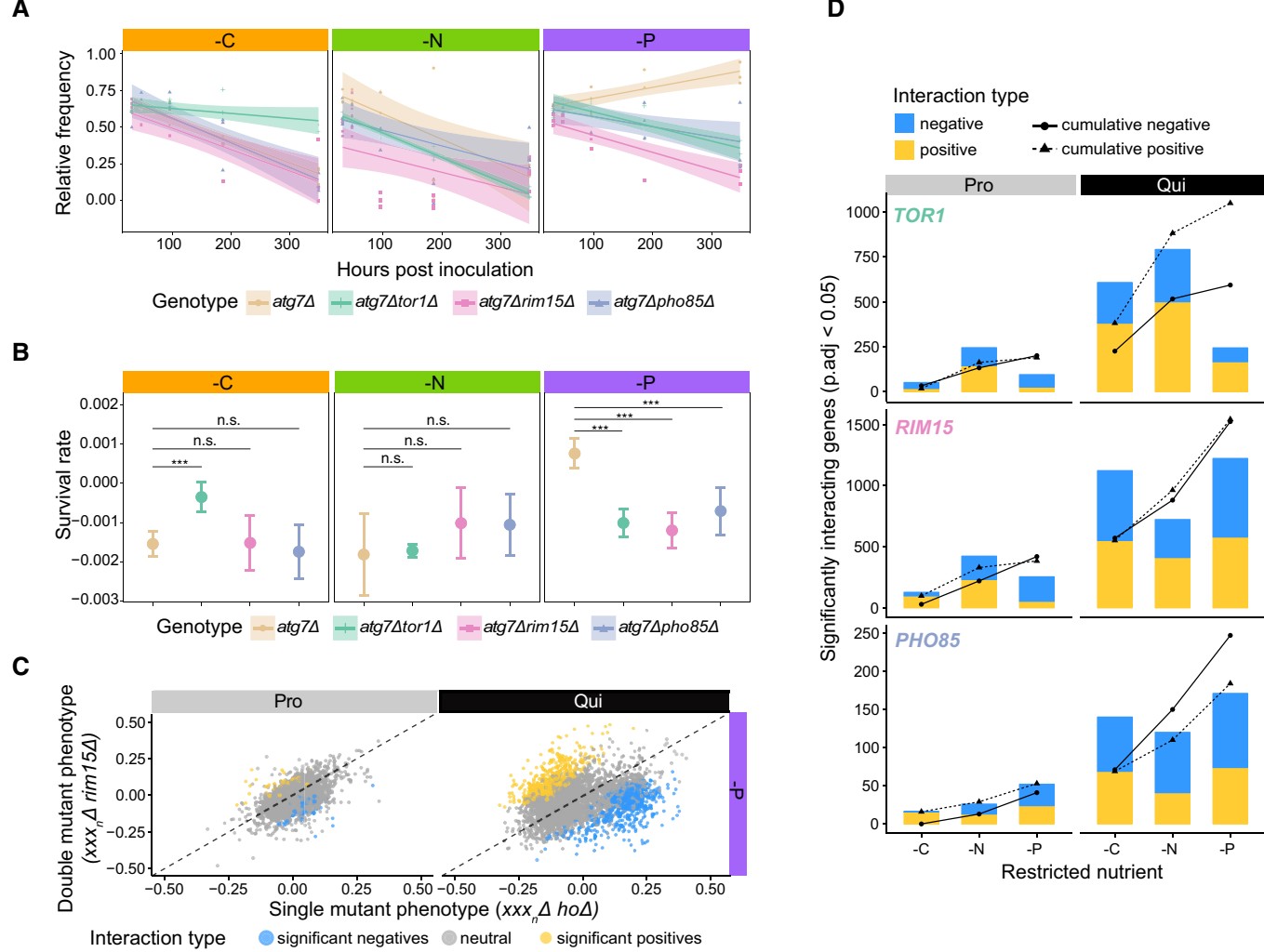

**Figure 3.  Identification of condition-specific genetic interactions using pooled double mutant analysis.**

A   Genetic interactions for each gene were determined for three different query genes (*TOR1*, *RIM15*, and *PHO85*) in three different conditions (-C, -N, and -P) and two different cellular states: quiescence (shown) and proliferation (not shown) using pooled mutant time series analysis.

B   Survival rate for each genotype indicated in A) and 95% confidence intervals. The false discovery rate (FDR) as set at 5% ($10 < n < 17$, ***$P_{adj} < 0.05$).

C   Relationship between single mutant phenotype ($xxx_n\Delta::natMX$) and the corresponding phenotype of the mutant in the background of a *RIM15* deletion ($rim15\Delta::KanMX\ xxx_n\Delta::natMX$) in two different cellular states (Pro—proliferation and Qui—quiescence). The dashed line is the line of equality. Blue dots are genes that show a significant negative interaction with *RIM15*, and yellow dots depict significant positive interactions.

D   At a false discovery rate (FDR) of 5%, different numbers of significant genetic interactions are detected for three regulatory kinases in the three nutrient restrictions and two cellular states. Solid lines with circles indicate the cumulative total number of unique negative interactions, and dashed lines with triangles indicate the cumulative total number of unique positive interactions.

indicating greater similarity among positive or negative interactions within each domain despite the relative paucity of significant interactions (Appendix Fig S7). This difference may reflect the fact that genetic interactions in quiescent cells reflect novel regulatory relationships compared with those identified using fitness measurements in rich media that were used to construct the reference map.

The functional annotation of genetic interactions for each kinase differs as a function of the cellular state. For example, functional domains related to respiration, oxidative phosphorylation, mitochondrial targeting, transcription, and chromatin organization are enriched for negative genetic interactions with *TOR1* and *PHO85* in

carbon-restricted proliferative cells (Fig 5), but we find no evidence for enrichment of these functions in quiescent cells starved for carbon (Fig 5). Similarly, in nitrogen-restricted conditions, *TOR1*, *RIM15*, and *PHO85* share similar coherent functional interactions in proliferative cells, which are not observed in quiescent cells starved for nitrogen.

In addition, the functional enrichment of genetic interactions for each kinase differs between the three different nutrient-restricted conditions. For example, ribosome biogenesis genes are enriched for negative interactions with *TOR1* in nitrogen-restricted proliferative cells (Fig 5), but in phosphorus-restricted proliferative cells,

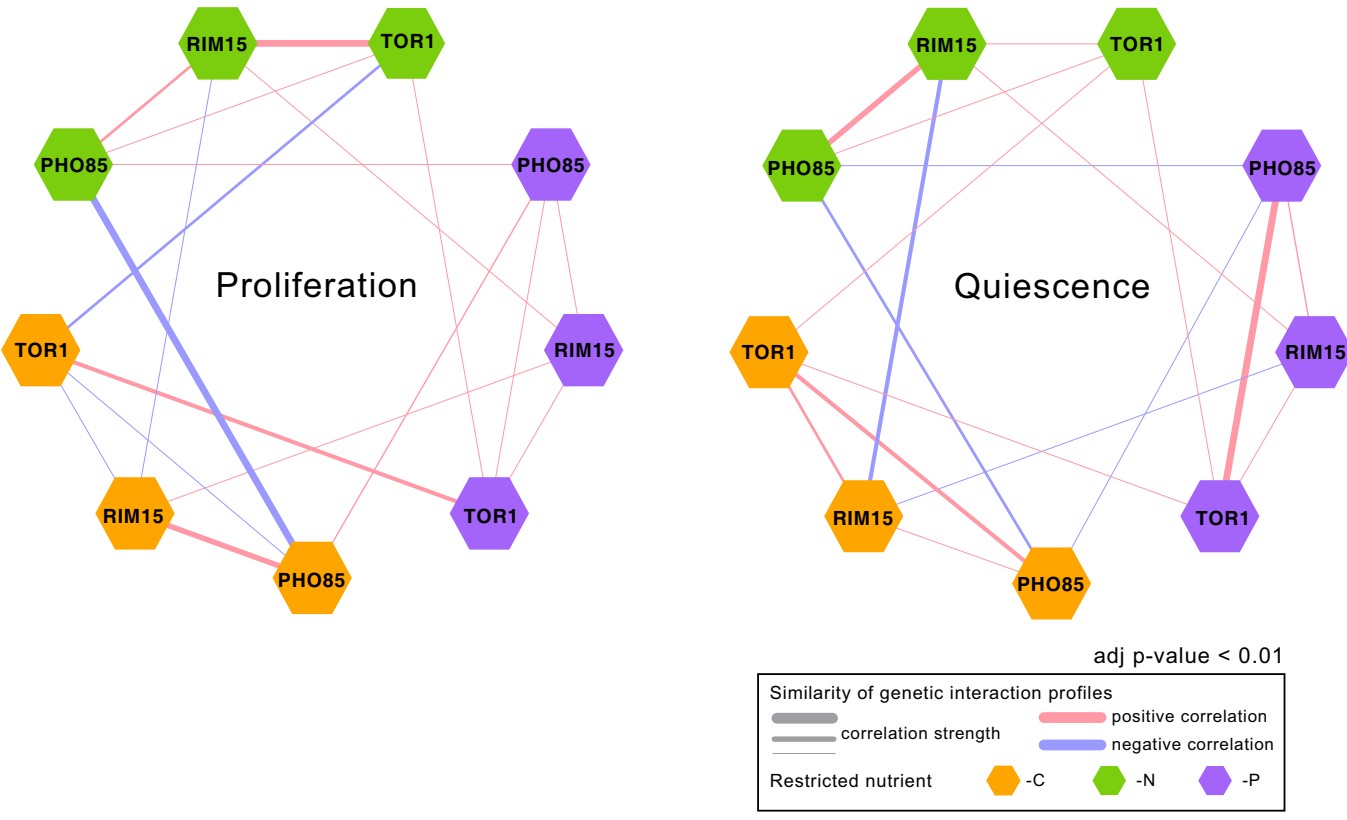

**Figure 4. Genetic interaction profile similarities are condition-dependent.**

Correlation networks based on genetic interaction profiles for *TOR1*, *RIM15*, and *PHO85* in proliferating cells (left) and quiescent cells (right) in three different nutrient-restricted media: carbon (-C), nitrogen (-N), and phosphorus (-P). Hexagons are color-coded based on the restricted nutrient type (orange for -C, green for -N, and purple for -P). Kinases with positive Pearson correlation are connected with pink edges, and kinases with negative Pearson correlation scores are connected with blue edges. The thickness of the edge indicates the strength of the correlation (i.e., a larger absolute correlation is represented by thicker edge).

ribosome biogenesis genes positively interact with *TOR1* (Fig EV5B). We find multiple additional cases of enrichment within functional domains, in which the sign of the genetic interactions is opposite between nitrogen and phosphorus restrictions in *TOR1* (Figs 5 and EV5B), suggesting that *TOR1* may play different regulatory roles in responding to nitrogen and phosphorus restriction.

We also found cases of functional enrichment that are maintained in the two different cellular states. For example, genes involved in peroxisome functions are enriched for negative interactions with *PHO85* in carbon-restricted proliferative cells and carbon-starved quiescent cells (Fig 5). This is consistent with the known role of *PHO85* in regulating long-chain-based kinase during stationary phase (Iwaki *et al*, 2005), suggesting that *PHO85* may play a role in maintaining long-chain fatty acid recycling and provide energy for cells in calorie-restricted conditions.

## Common and specific genetic interactions with *RIM15* support its role as a central mediator of quiescence

*RIM15* has previously been identified as an integrator of quiescence signals that is downstream of TOR1, PHO85, and PKA (Pedruzzi *et al*, 2003; Wanke *et al*, 2005; Olivares-Marin *et al*, 2018). Therefore, we expect that the genetic interaction profiles for *RIM15* should show more functional coherence in response to different quiescence

signals compared to *TOR1* and *PHO85*, which are upstream of *RIM15*. Using SAFE analysis, we find that *RIM15* consistently interacts with genes functioning in multivesicular body (MVB) sorting and pH-dependent signaling under all starvation conditions in both cell types (Fig 5). This suggests that *RIM15* plays an essential role in regulating protein homeostasis via MVB sorting. As the reference genetic interaction map used for SAFE does not include all genes in our genetic interaction dataset (only ~2,900 non-essential genes are present in the reference) and tests for coherence among both statistically significant and non-significant interactions, we performed overrepresentation analysis on the sets of genes that significantly interact with each kinase (Materials and Methods). Due to the limited number of significant interactions detected in proliferative cells (Figs 3D and EV3D), we did not find any enriched GO terms for kinases in proliferative cells. However, we identified multiple significantly enriched functional categories in quiescent cells. As with SAFE analysis, the functional enrichment of the significant interacting genes for a given kinase depends on the starvation signal (Figs 6A and EV6A).

Consistent with its role, *RIM15* genetic interactions in quiescent cells show more common functional enrichments in response to different starvation signals in comparison with *TOR1* and *PHO85*. Three functional groups are shared among genes interacting with *RIM15* in response to carbon/nitrogen or nitrogen/phosphorus

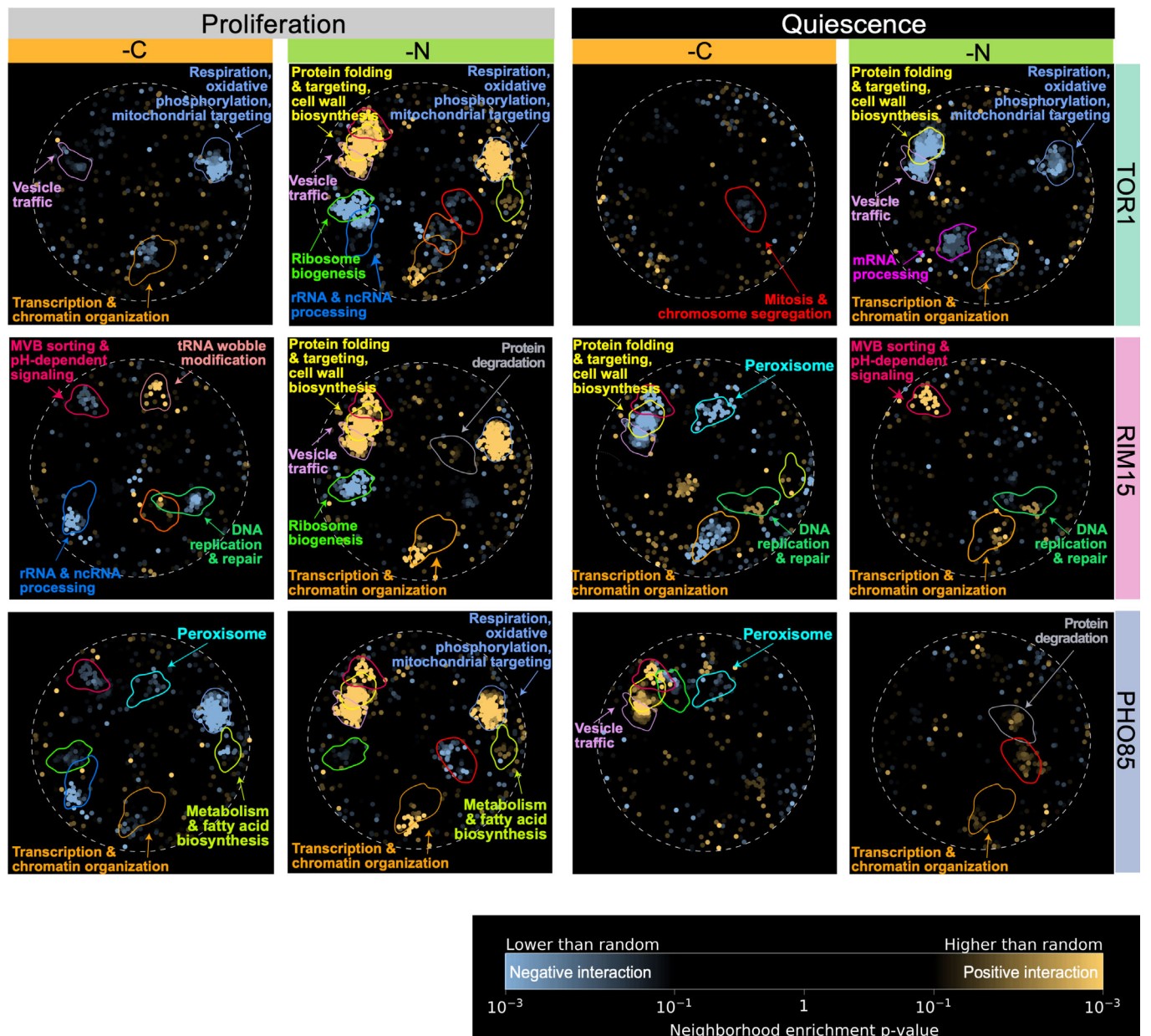

**Figure 5. Functional mapping of kinase genetic interaction profiles in proliferating and quiescent cells using SAFE.**

Genetic interaction enrichment landscape of *TOR1, RIM15, and PHO85* in proliferating and quiescent cells under different nutrient restrictions: carbon (-C) and nitrogen (-N). Each dot represents one gene. Blue dots represent genes that have negative interactions with corresponding kinase (row-wise) in each condition (column-wise), and yellow dots represent genes with positive interactions.

starvations (Fig 6A, lower panel), whereas there is limited or no functional overlap detected for *TOR1* or *PHO85* genetic interaction profiles under the same conditions (Fig EV6A and Table EV8). This is consistent with a model in which *RIM15* regulates quiescence through integration of diverse signals and execution of similar regulatory interactions. In quiescent cells, *RIM15* shows consistent genetic interactions with genes involved in vacuolar functions regardless of the starvation signals perhaps reflecting a role for *RIM15* in regulating autophagy and protein recycling in response to different starvations.

Interestingly, we find that genes that function in the endoplasmic reticulum-associated protein degradation, luminal domain monitored (ERAD-L) pathway show coherent positive interactions with *RIM15* specifically in nitrogen-starved quiescent cells (Fig 6A). This includes each of the genes that is known to function in ERAD-L: *USA1, YOS9, DFM1, HRD1, HRD3, CUE1, and DER1* (Figs 6B and EV6B). ERAD-L genes present in the genetic interaction reference data used for SAFE analysis; *HRD1, HRD3, CUE1, and USA1* are found in the domain enriched for ubiquitin-dependent protein catabolic process (Fig EV6C, red arrow). These results point to a

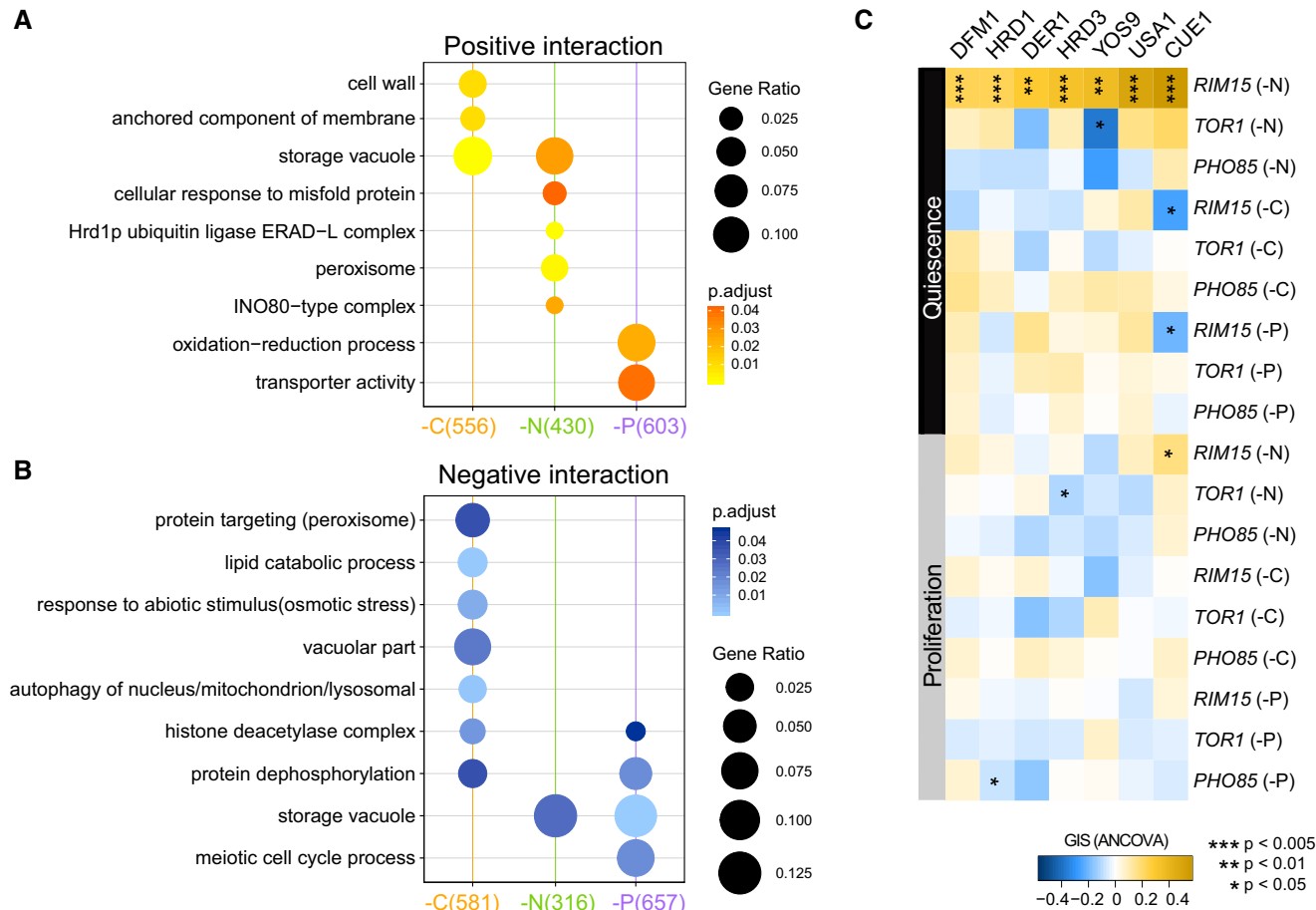

**Figure 6. *RIM15* genetic interaction profiles indicate its role as an integrator of quiescent signals with nutrient-specific functions.**

A  GO term enrichment analysis for genes that positively interact with *RIM15* in all nutrient starvation conditions. Only significantly enriched GO terms are shown ($P_{adj}$ < 0.05).

B  GO term enrichment analysis for genes that negatively interact with *RIM15* in all nutrient starvation conditions. Only significantly enriched GO terms are shown ($P_{adj}$ < 0.05).

C  Genetic interaction profile of the genes encoding the ERAD-L complex. ERAD-L genes show a unique cohesive set of positive genetic interactions with *RIM15* in nitrogen starvation-induced quiescent cells. Each column is the genetic interaction score between an ERAD-L gene and each of the kinase genes in each nutrient-restricted condition.

previously unknown specific function for *RIM15* in proteostasis regulation in response to nitrogen starvation via the ERAD-L pathway.

## Discussion

Cellular quiescence is the predominant state of eukaryotic cells. To study the genetic requirements of cellular quiescence in yeast cells, we quantified the effect of each gene deletion in response to three distinct nutrient starvation signals (carbon, nitrogen, and phosphorus). To study how these signals are coordinated within quiescent cells, we quantified genetic interactions with three regulatory kinases in each of the three starvation conditions. To undertake this study, we quantified phenotypic differences in different cellular states (proliferation versus quiescence) and genotypes (single versus double mutational background) using multiplexed barcoded

analysis to track thousands of different genotypes using time course analysis. By testing the contribution of ~4,000 yeast non-essential genes to fitness in proliferating cells and survival in quiescent cells in three different nutrient-restricted conditions, we find no evidence for genes that are commonly required for quiescence. We extended our method for multiplexed analysis of genotypes to study ~14,400 double mutants encompassing three core kinase genes: *TOR1*, *RIM15*, and *PHO85*, which allowed us to test for genome-wide genetic interactions with regulatory kinases that mediate quiescence.

The functional requirements for maintaining and exiting quiescence differ depending on starvation signals. Time course analysis of fitness during proliferation and survival during starvation supports previous findings that yeast cells have distinct functional requirements for maintaining viability of quiescent cells in response to different nutritional starvations (Klosinska *et al*, 2011). In addition, our results show that a substantial fraction of the non-essential

yeast genome is required for survival during quiescence independent of their requirements for growth. For example, in carbon-restricted conditions, deletion of 713 (~15%) of the non-essential genes results in a significant defect in quiescence (Fig 2C). Clearly, the definition of an "essential gene" is dependent on the condition in which essentiality is assessed.

Across all starvation conditions, we found that only 8 genes are commonly required for quiescence, a result that is not significantly different from chance (Fig EV2D). The absence of a common set of genetic requirements for quiescence in response to different natural nutrient starvation signals is consistent with earlier work (Klosinska et al, 2011). Although there appears to be no common set of genetic requirements for quiescence, different nutrient starvations do share some genetic requirements. Nitrogen- and phosphorus-starved quiescent cells tend to have more overlapping features than carbon starvation-induced quiescence: 81 genes are required for maintaining quiescence in response to both nitrogen and phosphorus starvation, whereas only 57 genes are commonly required for quiescence in nitrogen and carbon starvation (Fig 2C). Results from functional enrichment analysis are consistent with the trend of greater overlap in genetic requirements in nitrogen and phosphorus starvation. For example, genes involved in protein localization by CVT pathway are required in response to nitrogen or phosphorus starvation. The patterns of functional overlap in genetic requirements for responding to nitrogen, phosphorus, and carbon starvation may reflect their different primary biological uses: carbon is the major energy source, whereas both nitrogen and phosphorus are primarily utilized for macromolecular synthesis (Wilson & Roach, 2002; Broach, 2012; de Virgilio, 2012; Alberts et al, 2013).

To date, genome-wide genetic interaction mapping in yeast has primarily been assayed using a single phenotype in a single condition—colony growth in rich media. Our genome-wide genetic interaction mapping in different conditions and cellular states indicates that: (i) genetic interactions with regulatory kinases vary between conditions; (ii) genome-wide genetic interaction mapping is extensible to additional phenotypes and analyzing condition-specific phenotypes may increase the sensitivity for identifying novel regulatory relationships; (iii) less favorable conditions result in an increased number of significant interactions; and (iv) for a given physiological state (e.g., proliferation or quiescence), increasing the number of environmental conditions results in an increase in the number of significant genetic interactions. The points are consistent with, and extend, the limited number of studies that have investigated genetic interactions in different growth and stress conditions (St Onge et al, 2007; Gutin et al, 2015; Martin et al, 2015). Despite the fact that our genetic interaction dataset is limited in its scale and is focused on regulatory kinase genes, we anticipate that our methodology can be broadly applied to define genetic interactions in different conditions and cellular states.

Endoplasmic reticulum-associated protein degradation (ERAD) is a quality control mechanism that ensures only properly folded proteins leave the ER. Autophagy has been proposed to be a backup mechanism for ERAD. Previous studies have shown that RIM15 plays a role in regulating autophagy and protein homeostasis (Waliullah et al, 2017; Huang et al, 2018). In our study, we find that genes that function in ERAD show coherent positive interactions with RIM15 in nitrogen starvation conditions, suggesting that RIM15 regulation of ERAD activity in response to nitrogen starvation is

essential for quiescence. It is possible that RIM15 functions to regulate clearance of stress-induced misfolded proteins during nitrogen starvation by mediating the balance between autophagy and ERAD.

Our study has important implications for our understanding of the genotype to phenotype map. The prevailing result from our study is that the effect of a given gene deletion on a phenotype (either fitness or survival) is highly dependent on the specific environmental conditions of the cell. Although nitrogen, carbon, and phosphorus starvations all lead to cell cycle arrest and the initiation of quiescence, the genetic requirements for this behavior are distinct. We find that the conditional dependence extends to genetic interactions as we detect different sets of genetic interactions in different growth and starvation conditions. These results are consistent with our previous study of natural genetic variation in which we found that the effect sizes of QTL underlying fitness differences, and genetic interactions between QTL, are acutely sensitive to the composition of the growth media (Ziv et al, 2017). Identifying quantitative genetic effects and interactions that are insensitive to environmental variation appears challenging and may, in fact, be extremely rare.

It has been argued that starvation for glucose is the relevant condition for studying quiescence (Sagot & Laporte, 2019), and indeed, the vast majority of quiescence studies are performed in conditions in which carbon starvation is the pro-quiescence signal (Laporte et al, 2011, 2018). However, it has been appreciated for many decades that yeast cells can initiate a quiescent state in response to different starvation signals (Lillie & Pringle, 1980; Klosinska et al, 2011). Our study reiterates the importance of studying quiescence in response to different nutrient starvation conditions. Many important biological processes are likely to be missed— autophagy being a preeminent example—if carbon starvation is the only condition studied (Kawamata et al 2017). Organisms in the natural world experience a range of nutrient limitations, and nitrogen and phosphorus appear to be the predominant limiting nutrients in most ecologies (Elser et al, 2007). Thus, a complete understanding of cellular quiescence requires the study of different nutrient starvation signals.

The study of cellular quiescence may inform our understanding of cellular aging and provide insight into the therapeutic challenge of dormant cancer cells. Our study supports previous findings that quiescence establishment follows distinct routes depending on the nature of the inducing signal (Coller et al, 2006; Klosinska et al, 2011). In addition, different "degrees" of quiescence may exist (Coller et al, 2006; Gookin et al, 2017; Laporte et al, 2018) as we find that cells maintained longer in quiescence need more time to return to growth. Thus, quiescence may be viewed as a continuum that ultimately leads to senescence (even if that may take thousands of years) unless conditions favorable for proliferation are met.

Overall, our data highlight the fact that quiescence does not imply uniformity (O'Farrell, 2011). The idea that quiescence establishment is the result of a universal program is clearly an oversimplification. Our study points to a rich spectrum of condition-specific genetic interactions that underlie cellular fitness and survival across a diversity of conditions and introduces a generalizable framework for extending genome-wide genetic interaction mapping to diverse conditions and phenotypes. Deciphering the underlying regulatory rationale and the hierarchical relationships between these signaling pathways in different conditions is critical for understanding cellular quiescence.

# Materials and Methods

## Reagents and Tools table

| Reagent/Resource | Reference or source | Identifier or catalog number |
|---|---|---|
| **Chemicals, enzymes, and other reagents** | | |
| SYBR Green | Invitrogen | Cat # S7563 |
| LIVE/DEAD™FungaLight™ Yeast Viability Kit, for flow cytometry | Invitrogen | Cat # L34952 |
| **Software** | | |
| Cytoscape v3.7.1 | http://www.cytoscape.org | NA |
| metScape 3 Correlation Calculator v1.0.1 | http://metscape.ncibi.org/calculator.html | NA |
| Revigo | http://revigo.irb.hr/ | NA |
| **Other** | | |
| Kits, instrumentation, laboratory equipment | | |
| Illumina NextSeq 500 | Illumina | NA |
| PureLink™ Pro 96 Genomic DNA Purification Kit | Invitrogen | K182104A |
| Tecan Freedom Evo and Infinite Microplate Reader | Tecan | NA |
| Coulter counter | Beckman Coulter | NA |
| Flow cytometry | BD Accuri™ C6 | NA |
| QIAquick PCR purification columns | Qiagen | 28104 |
| Chemostats (Bioreactor) | | NA |

## Methods and Protocols

### SGA library construction

The haploid prototrophic double deletion collections were constructed using the synthetic genetic array method (Tong *et al*, 2001). The genotype and ploidy of double mutants are prototrophic haploid (Fig EV1B). For the single deletion collection (array mutants), gene deletion alleles are marked with the kanMX4 cassette conferring G418 resistance, which is flanked by two unique molecular barcodes (the UPTAG and DNTAG). For double deletion collection, an additional query allele is marked with NatR cassette conferring nourseothricin resistance. To construct the *RIM15* and *TOR1* SGA query strains, we mated a MATa $xxx_n\Delta0$::NATr strain (transformed from FY4 with a NATr PCR product targeting the $xxx_n$ allele) with the Y7092 strain. A haploid prototrophic strain was identified following tetrad dissection and genotyping using selective media with G418 and nourseothricin. To construct the *HO*, and *PHO85* SGA query strains, we transformed a prototrophic strain containing the SGA marker with a NATr PCR product targeting the *xxx* allele. Insertion of NATr was confirmed via PCR, and the genotype of the strain was checked via replica plating onto selective media resulting in strains listed in Table EV1.

### Growth conditions

After the growth of individual selected mutants on YPD agar plates, all mutants were pooled to a final density around $1.7 \times 10^9$ cells/ml. Each agar plate contained single colonies of individual genotypes and replicated colonies of the control *ho*Δ strain. We inoculated $1.5 \times 10^8$ cells into 300 ml of nutrient-limited medium: for glucose (C, 26.64 mM carbon), ammonia (N, 0.8 mM nitrogen), and phosphorus (P, 0.04 mM phosphorus) at 300 ml. To define the fitness of ~4,000 mutants we performed three-five independent experiments for each mutant per nutrient-limiting conditions. In total, we studied 4 mutant collections × 3-5 biological replicates × 3 nutrient limiting conditions in bioreactors maintained at 30°C and pH of 5. To determine the relative abundance of each genotype at different states spanning both proliferative and quiescent stages, we collected five time points in each stage (based on growth curve; Fig EV1C). The duration of the experiment was 15–16 days, and populations were sampled at 0, 9, 14, 18, 24, 32, 48, 96, 187, and 368 h for outgrowth and barcode sequencing. To isolate viable cells from the culture, we transferred 1 ml (i.e., $1 \times 10^6$ cells) from the pooled library at each time point into 5-ml minimal cultures. This sample was grown for 24–48 h to a final density of $3 \times 10^8$ cells/ml. Cells were then washed with water once, and then resuspended in 1 ml sorbitol buffer for genomic DNA purification.

### Viability quantification using propidium iodide and SYTO® 9

For viability quantification at each time point, $1 \times 10^7$ cells were collected and subsequently washed once with sterilized DI water and once with PBS. The washed cell pellet was resuspended with 1 ml 1× PBS and stained with 3.34 μM of SYTO® 9 and 20 μM of propidium iodide for 20 min. The stained samples were analyzed by flow cytometry (BD Accuri™ C6).

### DNA extraction and library preparation for Bar-seq

Genomic DNA was isolated from $1 \times 10^8$ cells for each sample (3 nutrient-restricted × 3-5 biological replicates × 4 deletion collections × 10 times points) using Invitrogen PureLink™ Pro 96 Genomic DNA Purification Kit. We used a two-step PCR protocol for efficient multiplexing of Bar-seq libraries (Robinson *et al*, 2014). Briefly, UPTAGs and DNTAGs were amplified separately from the same genomic DNA template. In the first PCR step, unique sample indices are added to each sample. For each biological replicate, we used 120 unique sample indices that differed by at least two nucleotides to label each sample from 3 nutrient-limiting conditions × 4 deletion collections × 10 time points. We normalized genomic DNA concentrations to 10 ng/ml and used 100 ng template-amplified barcodes using the following PCR program: 2 min at 98°C followed by 20 cycles of 10 s at 98°C, 10 s at 50°C, 10 s at 72°C, and a final extension step of 2 min at 72°C. PCR products were confirmed on 2% agarose gels, and the concentration was quantified using a SYBR Green staining followed by Tecan Freedom Evo and Infinite Microplate Reader. We combined 35 ng from each of the 120 different UPTAG libraries and, in a separate tube, 35 ng from each of the 120 different DNTAG libraries for each condition/deletion collection. The multiplexed UPTAG libraries were then amplified using the primers P5 (′5-A ATG ATA CGG CGA CCA CCG AGA TCT ACA CTC TTT CCC TAC ACG ACG CTC TTC CGA TCT-3′) and Illumina_UPkanMX, and the combined DNTAG libraries were amplified using the P5 and Illumina_DNkanMX primers using the identical PCR program as the first step with 75 ng template. The

140-bp UPTAG and DNTAG libraries were purified using QIAquick PCR purification columns, quantified using a Qubit fluorometer for qPCR quantification, combined in equimolar amounts after qPCR, and adjusted to a final concentration of 4 nM mixture of pooled UPTAG and DNTAG. In total, each sequencing library contained 120 UPTAG and 120 DNTAG libraries from 120 different samples. The library was sequenced on a single lane of an Illumina NextSeq 500 with High Output 1 × 75 bp read configuration. 20% PhiX was spiked into each library for increasing the complexity of two color base calling on the Illumina NextSeq500 platform.

### Data analysis, filtering, and normalization

Sequence reads were matched to the yeast deletion collection barcodes using reannotation by Smith et al (2009). Inexact matching was performed by identifying barcode sequences that were within a Levenshtein distance of 2 from each read (Levenshtein 1966). Sample indices were similarly matched using a maximum Levenshtein distance of 1. The final matrix of counts matching the UPTAG and DNTAG of each of the 480 samples is provided as Dataset EV7. 41 libraries with total read depth less than $1 \times 10^5$ reads were removed from the 960 libraries. We merged the UPtag and DOWNtag counts representing the same gene within each condition resulting in 439 libraries in total. A set of outliers was identified that had fewer than 3,000 total reads across all 439 samples. These low-count matches (=< 4) were likely due to sequencing errors and were removed. 1,996 mutants were removed with a coverage < 3,000 or missing in either tag counts. After filtering, a matrix containing 3,931 mutants consistent with high-quality counts data across conditions was generated corresponding to 857,016,574 sequence reads. This counts table was normalized using the function varianceStabilizingTransformation in the DESeq2 package (Love et al, 2014) (version 1.8.1) with arguments blind = FALSE and fitType = "local".

### Fitness, survival, and phenotypic difference quantification

The normalized frequency of each mutant within each library was used for linear regression modeling. For example, in the HO library, the count of each mutant (ho::kanMX xxx_n::natMX) is normalized by the count of the wild-type control (hoΔ::kanMX his3Δ1 can1Δ::STE2pr-Sp_his5) at each same time point. In the other double mutant libraries, the counts for each double mutant (query::kanMX xxx_n::natMX) is normalized by the counts of the query mutant (queryΔ::kanMX his3Δ1 can1Δ::STE2pr-Sp_his5) (Dataset EV8). For each mutant strain N, fitness $f_n$ was calculated as the coefficient of a linear regression model using R:

$$lm\left(\frac{F^n}{F^{wt}} \sim T\right),$$

$F^n$ is the count of strain N at each time point, and $F^{wt}$ is the count of the wildtype or query mutant strain at the same time point. T refers to time points, which was measured in days for quantifying the fitness in prolonged starvation. $\delta$ is the error term.

To compare the phenotypic difference for a given mutant between different cellular states, before linear regression modeling we scaled the independent variable, time (hours) for each stage into the same unit maintaining the natural interval using the scale() function in R with center = FALSE. For example, the time point (independent variable) in the proliferative stage was scaled from 0 h, 9 h, 14 h, 18 h,

and 24 h into 0, 0.5246676, 0.8161497, 1.0493353, and 1.3991137, and the time point for sample collected during quiescence was scaled from 32, 48, 96, 187, and 368 h into 0.1553874, 0.2330811, 0.4661622, 0.9031894, and 1.6995499. Then, we quantified the phenotypic difference between fitness in proliferation and survival in quiescence using ANCOVA:

$$lm\left(\frac{F^n}{F^{wt}} \sim T * GS\right)$$

where T is the scaled time, and GS is the Growth Stage (e.g., proliferation or quiescence). The different growth stages in this function is the interaction term, which was used to test for statistical significance.

After quantifying the phenotypic difference between quiescence and proliferation for a given mutant, we ranked the mutants by phenotypic difference in a descending order and then applied Gene Set Enrichment Analysis (GSEA) using clusterProfiler (Yu et al, 2012).

To test whether the common genes required in response to different quiescent signals are statistically significant we used the multiset intersection test algorithm in the R software package SuperExactTest (Wang et al, 2015).

### Comparison of SGA genetic interaction quantification with ANCOVA

*SGA genetic interactions scoring method*

We first computed genetic interactions using a method analogous to estimation of epsilon (ε) as defined in SGA screens. The SGA-like score was quantified by testing the null hypothesis based on a multiplicative model from single mutant fitness: $\varepsilon = f_{aq} - f_a * f_q$ (a —single array mutant; q—single query mutant, aq—double mutant).

In our case, ε is calculated as the difference between the coefficients of linear modeling:

where

$f_{aq}$ is the coefficient generated by $lm\left(\frac{F^{aq}}{F^q} \sim T\right)$,

$f_a$ is the coefficient generated by $lm\left(\frac{F^a}{F^{wt}} \sim T\right)$,

$f_q$ is the coefficient generated by $lm\left(\frac{F^q}{F^{wt}} \sim T\right)$,

$f_{aq}$, $f_a$, and $f_q$ are normally distributed around 0 with positive (better than WT) and negative (worse than WT) phenotypic readouts (fitness in proliferative cells and survival in quiescent cells). To estimate the expected fitness/survival in double mutant based on multiplicative model, we take the exp() of the coefficients for each model to eliminate the discordance of the signs in fitness/survival. Then, we calculated the expected fitness/survival using multiplicative model:

$$f_{aq}^{exp} = exp\ (f_q) \times exp\ (f_a)$$

Therefore,

$$\varepsilon = exp\ (f_{aq}) - f_{aq}^{exp}$$

$S_a$ and $S_q$ are the standard errors of each linear model. The standard error in expected fitness/survival is calculated by propagating standard error from each individual model:

$$S_{a+q}^2 = S_a^2 + S_q^2$$

Then, the statistical significance between expected (multiplicative model) and observed model was calculated by Welch's *t*-test:

$$t = \frac{f_{aq} - f_{a+q}}{\sqrt{\frac{S_{aq}^2}{N_{aq}} + \frac{S_{a+q}^2}{N_{a+q}}}}$$

where the degrees of freedom associated with this variance estimate is approximated using the Welch–Satterthwaite equation:

$$\upsilon \approx \frac{\left(\frac{S_{aq}^2}{N_{aq}} + \frac{S_{a+q}^2}{N_{a+q}}\right)^2}{\frac{S_{aq}^4}{N_{aq}^2(N_{aq}-1)} + \frac{S_{a+q}^4}{N_{a+q}^2(N_{a+q}-1)}}$$

*Genetic interaction quantification by using ANCOVA*
All libraries were normalized by the common query deletion, which has the effect of accounting for a global general effect on fitness/survival that is attributable to the query allele. Therefore, the GIS can be calculated by looking at the difference between normalized fitness/survival without incorporating the single query mutant phenotype,

$$GIS = f_{aq} - f_a,$$

where

$$f_{aq} = lm\left(\frac{F^{aq}}{F^q} \sim T\right) \,\&\, f_a = lm\left(\frac{F^a}{F^{wt}} \sim T\right),$$

In this case, the genetic interaction is calculated directly by testing whether the query mutation significantly changes the relationship between time and relative fitness for a given mutant. We applied ANCOVA using:

$$lm\left(F^{normed} \sim T * GT\right)$$

where $F^{normed}$ is the query allele normalized frequency, e.g., $F^a/F^{wt}$, $F^{aq}/F^q$. $T$ is the scaled time, and $GT$ is the Genotype (e.g., *HO*, *RIM15*). The significance of the interaction term was determined using a *t*-test.

### Functional annotation with clusterProfiler
Gene Set Enrichment Analysis (GSEA) was applied on the ranked gene list based on phenotypic difference using clusterProfiler (Yu *et al*, 2012). The GO overrepresentation test was applied to significantly interacting genes and quiescent specific gene lists using clusterProfiler.

### Network Construction using Cytoscape 3.7.1
The correlation among genetic interaction profiles was calculated by metScape 3 Correlation Calculator v1.0.1 using the DSPC method and then visualized in Cytoscape 3.7.1.

### Spatial analysis of functional enrichment (SAFE)
For SAFE, we used all genes without filtering based on statistical interaction significance (from ANCOVA). The visualization and local enrichment annotation was performed according to (Baryshnikova, 2016).

**Expanded View** for this article is available online.

## Acknowledgements
We thank Charlie Boone and Michael Costanzo for library construction. We thank the members of the Gresham Lab and Vogel Lab for helpful discussions. We thank NYU Gencore for next-generation sequencing and FACS. This work was supported by grants from the NIH (R01 GM107466) and NSF (MCB1818234).

## Author contributions
DG and SS designed the study; SS and NB performed the experiments; SS analyzed the data; AB provided the code for SAFE analysis; and DG and SS wrote the manuscript.

## Conflict of interest
The authors declare that they have no conflict of interest.

## Data availability

The datasets and computer code produced in this study are available in the following databases:
(i) All fitness, survival, and genetic interaction data: Open Science Frame (OSF): https://osf.io/6avpn/
(ii) Analysis code: GitHub https://github.com/ss6025/GI-of-kinases-in-quiescence
(iii) Interactive data http://shiny.bio.nyu.edu/ss6025/shiny_Genetic_Interaction/
(iv) Fastq files: Sequence read archive [SRP217852]: https://trace.ncbi.nlm.nih.gov/Traces/sra/?study=SRP217852

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
