## [Review Process File · Molecular Systems Biology]

Kinase genetic interaction profiles differ between environmental conditions and cellular states

David Gresham, Siyu Sun, Nathan Brandt, and Anastasia Baryshnikova

Review timeline:

Submission date:	14 th August 2019
Editorial Decision:	20 th September 2019
Revision received:	19 th December 2019
Editorial Decision:	17 th February 2020
Revision received:	18 th March 2020
Accepted:	31 st March 2020

Editor: Maria Polychronidou

Transaction Report:

1st Editorial Decision

20th September 2019

Thank you again for submitting your work to Molecular Systems Biology. We have now heard back from the three referees who agreed to evaluate your study. As you will see below, the reviewers acknowledge that the addressed topic is timely and find the conclusions potentially interesting. They raise however a series of concerns, which we would ask you to address in a revision.

I think that the reviewers' recommendations are rather clear and there is therefore no need to repeat the comments listed below. Please feel free to contact me in case you would like to discuss in further detail any of the issues raised by the reviewers.

REFeree REPORTS

Reviewer #1:

In their manuscript 'Regulatory kinase genetic interaction profiles differ between environmental conditions and cellular stress', Sun and colleagues investigate genetic interaction with kinases in yeast across different nutrient-restricted conditions. They establish a comprehensive screening approach to test all non-essential gene interactions with three kinases TOR1, RIM15 and PHO85 required for cell cycle progression and measure interactions in the quiescent and proliferating cell population. They identify several-fold more interactions in quiescent cells, suggesting that this distinction is valuable for assigning cell cycle phase-specific interactions. They then explore how genetic interactions of the three kinases differ between cell populations and growth conditions. The focus of the paper is timely as it addresses the important question of genetic dependency networks respond to dynamic environments, and the experimental set-up seems robust and well-executed. However, the analysis of the data does not continue beyond general exploratory efforts,

some of which are incomplete. Some of the claims made aren't well-supported by the analysis. Overall, this manuscript has substantial potential but could be significantly improved (see specific concerns/suggestions below).

Major concerns:

1. Defining a more precise phenotype than general fitness helps to draw better defined biological conclusion from the data. However, how the quiescent and proliferating population relate to each other is not well explored. For instance, the authors might want to establish to what degree those two populations complement each other. This would be crucial to understand what fraction of the more general cell fitness phenotype is composed of either of those populations. Importantly, this would allow a more biological interpretation of the genetic interaction data.
2. It would also be important to relate the phenotype signatures to the cell fitness signature of the same three kinases in the global reference map (Costanzo et al., 2016), since this work currently serves as gold standard. It is unclear the extent to which this manuscript improves existing data.
3. Media conditions are an important and very interesting part of this manuscript. However, the authors fail to establish an expectation (i.e. specific genes interacting with a given kinase in quiescent or proliferating cells) and show that this is indeed what they observe. This would help to build trust in biological interpretations of each condition generated. For instance, showing that specific genes act in a given condition according to their known biological function would help.
4. The authors explore two different models to identify genetic interactions and chose ANCOVA. To be able to judge those models, it will be helpful to see model fits, enrichments for known relations (e.g. GO terms, PPI). For instance, a fit of double vs single mutant scores is shown in Figure 3C, and it would be helpful to understand how the different models look like.
5. Figure 5 presents clear patterns / local functional enrichments particularly for the proliferating cell population. However, Figure 3D shows that many more genetic interactions are identified in quiescent cells. How do those two observations fit? Can the authors show that the large number of genetic interactions in quiescent cells are biologically relevant?
6. The authors claim to investigate signaling pathways governing cell cycle progression, stating that there is limited or no knowledge and/or experimental efforts to address this. In fact, the first study to systematically identify context-specific genetic interactions in yeast was published by St Onge, Nat Genetics 2006 (not cited). Later work has shown that cell cycle-specific genetic interactions reconstruct gene-gene networks relevant for cell cycle progression (Fischer et al., eLife 2015 and Billmann, Horn et al., MBoC 2016) as well as the fact that genetic interactions in signaling pathways largely rewire in response to stimuli (Billmann et al., Cell Systems 2018). By placing their work in the context of those studies, the relevance and novel insights of this study would become more evident.

Minor concerns:

1. The different influence of nutrient limitation and genetic background should be analyzed more directly (with regard to Figure 1B). Generalizing this concept will be important for the field but this manuscript currently does not seem to test for that specifically.
2. The opposite fitness profile of carbon-restricted screens (with regard to Figure 1B) as compared to the other two media conditions is not immediately clear when looking at the heatmap in Figure 1B. The authors could visually support this statement by providing direct comparisons (e.g. scatter plots).
3. The authors mention both 3,600 genes and 4,800 strains on the array used. Please clarify how those numbers relate.
4. In the legend for Figure 1, descriptions for panels B and C are mixed up.

Reviewer #2:

Summary and General Remarks:

The authors perform a genome-wide genetic interaction assay with three genes mediating quiescence. By using an informative phenotype for these genes, they find an enrichment of genetic

interactions, and are better able to study the biology of cell quiescence. I thought that the two most interesting findings were that: 1) many genes that are dispensable for proliferative growth are required for quiescence, and 2) there is lack of common quiescence-specific genes. Overall, I think this is a well-executed study and that the findings will be interesting to the readership of Molecular Systems Biology, and I recommend only minor revisions.

Major points:

My most pressing comment is on their finding that quiescence results in a five-fold enrichment of genetic interactions. It's previously been shown that genetic interactions are more common within a set of genes which are related to the phenotype being studied - e.g. St Onge et al 2007 find that when making double-knockout pairs of 26 genes that exhibit single-deletion sensitivity in MMS, 35% of gene pairs also have a genetic interaction in that condition. So, this enrichment may not be unprecedented. However, the authors also state that single-mutant effects predict genetic interaction propensity, and that quiescence results in increased variance in the distributions of survival compared to the distributions of fitness. While a greater interaction propensity amongst genes with greater single-mutant effects is likely to be at least partly biologically meaningful, it may also suggest that it is more difficult to model expected fitness when combining two single mutants with sizeable fitness defects. Thus, I am concerned that some GI enrichment may result from general deviations from either the GIS or the ANCOVA model (which are very similar) when combining large-effect mutations. Perhaps it may be possible for the authors to show that combining larger single-gene effects doesn't generally result in more deviations. If it does, they may control for the distribution of single-gene effects to see if an enrichment of interactions would still be present, or use a more empirical model for the combined effects of multiple mutations.

Minor points:

- Figure 1C (right panel) could use a more informative x-axis title than 'Coefficient of lm '
- Captions for Figure 1B and C are swapped
- Figure 2A is hard to read, I suggest the shading be removed
- Figure EV2B: why do many more mutants result in better fitness in proliferation than result in defects? (43-50% vs 8-12%). This does not seem consistent with the violin plot in Figure 2B

Reviewer #3:

Sun et al present a series of well designed experiments to investigate the effects gene deletions on growth and quiescence under different nutrient limitations, and the role of epistasis in these biological processes.

The paper is scholarly and well-written, and I enjoyed reading it, and there are some interesting results in there. I do have a number of questions, largely to get clarification on some things I didn't quite understand, and there are also some issues that I enumerate below.

Important things:

I was pretty shocked that some of the experiments did not cluster by medium - namely the TOR1 library in nitrogen, and the RIM15 in phosphorous. What makes it more surprising is that the TOR1 library in nitrogen looks pretty similar to the other libraries in Phosphorous, and RIM15 in phosphorous looks like the other libraries in carbon. I can wrap my head around that maybe they would look different, but having them cluster with libraries from another condition does raise the specter of there being some samples either misidentified/mixed up, or something funky happened to the media. Is there any way of post-facto determining that that couldn't be the case?

It struck me as odd to have the data clustered in Fig 1B, but for the paper to then say that it was much more useful (and I agree) to separate the time course into two pieces, to look at fitness during growth, vs. fitness during stationary phase, but that clustering of these data was not done - e.g. each deletion would have two values where it previously had one. I think clustering those data would be more useful than the combined data. Also, the contrast in Fig 1 was such that it's hard to see stuff. I think changing the contrast would help.

With regards to the experimental design, I do have concerns about the effects of drift and counting noise in assaying the abundance of the strains as they go through stationary phase. The paper says that they removed 2×10^5 cells for expansion to count relative abundance. Given the number of strains in the pool, this is only ~ 40 /barcode, if they start out evenly, which is somewhat low. However, I think this gets exacerbated as time goes on, because viability drops during starvation - did they assess viability to know that the number of viable cells being plated was sufficient to avoid drift and counting noise?

I'm presuming that the entire section "Distinct cellular functions...", and the data in Figure 2, are based on just the control data in the HO background, but it should be made explicit (if it's not the case, I would like to know why).

The authors say that many of the genes important for quiescence show no fitness effects in proliferation, but I think I disagree. RGT2 and GPR1 seem to show profound fitness increases during the proliferative phase, suggestive of a trade-off. The same is true (if not more so) for the genes involved in phosphor limitation, where those pointed out appear to be the most fit, including two other RIM genes, so maybe I misunderstood something.

I didn't really understand why the authors chose as the interesting genes in quiescence those that have a difference between their quiescent and proliferative fitness (Squi - Fpro) - why isn't just survival (or lack thereof) the important quantity? It seems like this will change the analysis substantially, and thus the conclusions, so I want more of a rationale as to why it isn't simply sufficient to ask which survive the worst during stationary phase.

For determining Squi, did the authors use a different starting timepoint for the carbon limitation, given that growth persists for longer? It seems like that should be the case.

Minor issues:

Appendix_Fig1 does not exist - I looked through the uploaded files, but couldn't find it

The graph in Figure 1 has an x-axis labeled as "hours post nutrient restriction", but I would prefer as "hours post inoculation"

I think it's pretty well known that genes involved in the Ras pathway improve fitness during growth under carbon limitation, but that they exhibit severe trade-offs in stationary phase, so earlier work to that effect should be cited (I can find references if the authors are unable to do so)

Figure 5 is pretty hard to read, especially the key. Please play with the color scheme to come up with something more interpretable.

The data in EV4A are pretty cool/interesting, but I'd also like to see the actual clustered data (maybe with some zoom ins to interesting groups of genes).

Trivial points:

In the Introduction, "ethanol as carbon source" should be "ethanol as the carbon source"

In the Results, "mutants was used to inoculate" should be "mutants were used to inoculate", as were refers to "cells"

In Figure 1, the legends for B and C are switched (or their labeling on the Figure is switched)

Gene names in the Methods should be italicized

*Please find below our response to each point raised by the three reviewers. Our responses are indicated in **red** and major changes made to the text are excerpted here and indicated in **bold**.*

Reviewer #1:

In their manuscript 'Regulatory kinase genetic interaction profiles differ between environmental conditions and cellular stress', Sun and colleagues investigate genetic interaction with kinases in yeast across different nutrient-restricted conditions. They establish a comprehensive screening approach to test all non-essential gene interactions with three kinases TOR1, RIM15 and PHO85 required for cell cycle progression and measure interactions in the quiescent and proliferating cell population. They identify several-fold more interactions in quiescent cells, suggesting that this distinction is valuable for assigning cell cycle phase-specific interactions. They then explore how genetic interactions of the three kinases differ between cell populations and growth conditions.

The focus of the paper is timely as it addresses the important question of genetic dependency networks respond to dynamic environments, and the experimental set-up seems robust and well-executed. However, the analysis of the data does not continue beyond general exploratory efforts, some of which are incomplete. Some of the claims made aren't well-supported by the analysis. Overall, this manuscript has substantial potential but could be significantly improved (see specific concerns/suggestions below).

We thank the reviewer for recognizing the significance of our study and for their constructive comments regarding the data analysis. We have addressed these comments through further analysis and modifications to the text as detailed below. We believe the paper is much improved as a result of these changes.

Major concerns:

1. Defining a more precise phenotype than general fitness helps to draw better defined biological conclusion from the data. However, how the quiescent and proliferating population relate to each other is not well explored. For instance, the authors might want to establish to what degree those two populations complement each other. This would be crucial to understand what fraction of the more general cell fitness phenotype is composed of either of those populations. Importantly, this would allow a more biological interpretation of the genetic interaction data.

The motivating rationale for our study was that most microbial cells spend the majority of their life in a non-proliferative state. Fitness is the ability to survive and reproduce - if most of a cell's life is spent in a non-proliferative quiescent state it follows that this must be a major component of fitness. In this study we have analyzed these two components of fitness separately for the first time.

To address this comment, we have made several changes throughout the text to make this point clearer in our introduction, our results, and in our interpretation of the differences between single gene and double mutant effects in proliferative and quiescent cells. We addressed this point specifically in our results section on page 11 stating:

To test whether the genetic requirements for proliferation in nutrient-restricted media and quiescence in response to starvation for the same nutrient are distinct, we investigated the fitness and survival of each genotype in the single mutant library (i.e. the *HO* library). We find that fitness in proliferation and survival in quiescence are poorly correlated for all three nutrient-restricted media (Fig EV3A).

2. It would also be important to relate the phenotype signatures to the cell fitness signature of the same three kinases in the global reference map (Costanzo et al., 2016), since this work currently serves as the gold standard. It is unclear the extent to which this manuscript improves existing data.

Comparison of our fitness measurements to the global reference map is an excellent idea. To address this comment, we extracted the data for *RIM15*, *TOR1*, *PHO85* from Costanzo et al. 2016 and compared it with our data for the three proliferative and three quiescent conditions. We compared both fitness profiles and genetic interactions. We find very poor correlation with genome-wide fitness profiles and genome-wide genetic interaction profiles.

To test whether this reflects different sources of noise in the two assays, we considered only significant genetic interactions that were identified in Costanzo et al., 2016 and in our data. We find that there is poor agreement, even for genetic interactions identified in our proliferative conditions. Finally, we performed functional enrichment analysis using SAFE, which also shows no correlation between our data generated in the most analogous condition (cells growing in carbon) and the data from Costanzo et al 2016.

We believe that the fact that we do not reproduce the genetic interactions identified in Costanzo et al., is due to the fact that we are looking specifically at regulatory kinases that respond to nutrient availability. As our conditions (nutrient limitation and nutrient starvation) are radically different to those used by Costanzo et al (rich undefined medium) we expect that the activity of the encoded gene products is likely to be very different. This may be different for genes that do not encode products whose activity is so dependent on the external environment. We have added the results of this analysis and our interpretation to the revised manuscript on page 15.

To investigate the utility of using additional phenotypes in genetic interaction mapping, we compared both fitness estimates and genetic interactions identified in our study with the global reference set (Costanzo et al. 2016). As our conditions (nutrient limitation and nutrient starvation) differ substantially from those used in the global reference set (rich undefined medium) the genetic requirements are likely to be distinct. As expected, no significant

correlation was detected for fitness measurements in all conditions (three proliferative and three quiescent conditions) or kinases (Fig EV5), supporting the notion that fitness effects are highly conditionally dependent. Similarly, no significant correlation was detected between the genetic interaction profiles quantified in both studies (Fig EV6). To test whether this reflects different sources of noise in two assays, we considered only those significant genetic interactions that were identified in both Costanzo et al., 2016 and our study. We find that there is poor agreement, even for genetic interactions identified in our carbon-restricted proliferative conditions, which is the most analogous condition to rich undefined media (Fig EV6).

3. Media conditions are an important and very interesting part of this manuscript. However, the authors fail to establish an expectation (i.e. specific genes interacting with a given kinase in quiescent or proliferating cells) and show that this is indeed what they observe. This would help to build trust in biological interpretations of each condition generated. For instance, showing that specific genes act in a given condition according to their known biological function would help.

This is an excellent point. We now provide additional text stating our expectations regarding which single gene deletions are likely to be deleterious in each of the conditions and the extent to which those expectations are met (page 12). We have not added an equivalent statement for genetic interactions as we had no a priori expectation of which genes should interact with a given regulatory kinase gene.

We identified significantly enriched GO terms ($p_{\text{adj}} < 0.05$) and find that functions involved in responding to the specific starvation signal are required for survival. For example, trehalose accumulation provides a reserve of fermentable sugar to reinitiate the cell cycle and provides protection against stress in quiescence (Gray et al. 2004; Shi et al. 2010; Klosinska et al. 2011). Therefore, we expect to see mutants defective in trehalose storage should fail to survive when starved for carbon. Indeed, this is the case, but the impairment of this function does not impact survival when starved for nitrogen or phosphorus (Fig 2B & Table EV7). Autophagy has previously found to affect survival during phosphorus starvation (Gresham et al. 2011), which has also been recapitulated in our assay (Fig 2B). Similarly, our observation that genes required for survival of nitrogen starvation are uniquely enriched for selective autophagy of nucleus related amino acid trafficking and recycling (Fig 2B) is consistent with protein degradation involving autophagy playing a major role in nitrogen recycling (Tesnière, Brice, and Blondin 2015).

4. The authors explore two different models to identify genetic interactions and chose ANCOVA. To be able to judge those models, it will be helpful to see model fits, enrichments for known relations (e.g. GO terms, PPI). For instance, a fit of double vs single mutant scores is shown in Figure 3C, and it would be helpful to understand how the different models look like.

We have now performed a thorough comparison of genetic interaction scores measured using the two approaches, which is now summarized in Figure EV4A. We also performed functional enrichment analysis for genetic interaction profiles

generated using both methods (Table EV12). The results are largely equivalent and we now discuss this in the text reiterating the value of using ANCOVA. We have also added additional interpretations in the results section (page 17)

We find that the agreement between the two approaches is high (pearson's $R > 0.9$) when applied to both fitness in proliferative cells and survival in quiescent cells. The genetic interaction profiles calculated by ANCOVA (Table EV9) and the multiplicative model for both *TOR1* (Fig EV4A & Table EV10) and *RIM15* (Fig EV4B & Table EV11) are highly correlated across all nutrient-restricted conditions. As the *PHO85* deletion allele was not identified in the single mutant library (possibly due to an erroneous barcode) we could not perform this comparison for these genetic interactions. To further compare the two approaches, we applied GSEA to genetic interaction profiles calculated using each model and compared the similarity of generated GO terms using GoSemSim (Yu et al. 2010). The significant GO terms for a given condition identified using the different models are very similar (Table EV12), indicating that ANCOVA identifies the same genetic interactions and functional groups as the classic multiplicative model. As ANCOVA has a well developed statistical framework for error estimation and significance testing, we elected to use ANCOVA to compute GIS for all subsequent analyses.

5. Figure 5 presents clear patterns / local functional enrichments particularly for the proliferating cell population. However, Figure 3D shows that many more genetic interactions are identified in quiescent cells. How do those two observations fit? Can the authors show that the large number of genetic interactions in quiescent cells are biologically relevant?

That is a valid observation. There are a few reasons why we observe an apparent discrepancy between the number of interactions and the strength of the functional enrichment using SAFE: 1) Figure 3D plots the number of significant interactions, whereas Figure 5 uses all interactions (significant and non-significant) for calculating enrichment. The discrepancy might indicate that even weak (i.e., not strictly significant) interactions are non-randomly distributed throughout the network and produce a strong enrichment. Being able to analyze all interactions is one of the reasons SAFE was developed as the threshold for statistical significance is arbitrary, 2) It is possible that, for some of the screens, more interactions mean more noise, although that is clearly not the rule (e.g., *RIM15* on carbon-restricted condition has more interactions and better enrichment in quiescent cells than that in proliferative cells, whereas *TOR1* on carbon-restricted condition has more interactions but less enrichment in quiescent cells than in proliferative cells). We added the following text on page 18 to explain this:

Importantly, SAFE uses the entire set of genetic interactions for a given query genes, including those interactions that do not reach statistical significance, which allows identification of functional enrichment trends that may exist despite a lack of significance associated with each individual genetic interaction. We superimposed genetic interaction profiles of each kinase in each of the three nutrient-restricted media and both cellular states (Table EV14) onto the reference network using SAFE. We find that kinases that show

higher similarity in genetic interaction profiles (Fig 4) also show more similar enrichment patterns using SAFE analysis (Fig 5).

6. The authors claim to investigate signaling pathways governing cell cycle progression, stating that there is limited or no knowledge and/or experimental efforts to address this. In fact, the first study to systematically identify context-specific genetic interactions in yeast was published by St Onge, Nat Genetics 2006 (not cited) (St Onge et al. 2007). Later work has shown that cell cycle-specific genetic interactions reconstruct gene-gene networks relevant for cell cycle progression (Fischer et al., eLife 2015 and Billmann, Horn et al., MBoC 2016) as well as the fact that genetic interactions in signaling pathways largely rewire in response to stimuli (Billmann et al., Cell Systems 2018). By placing their work in the context of those studies, the relevance and novel insights of this study would become more evident.

We thank the reviewer for drawing our attention to these important references. We have included all of these suggested references in our introduction and discussion better placing our results in the context of previous work.

Page 5: Quantitative genetic interaction mapping is increasingly being applied in other organisms, including *Drosophila melanogaster* and mammalian cells using RNAi or CRISPR (Fischer et al. 2015; Billmann et al. 2016, 2018; Du et al. 2017) making these questions of broad significance.

Page 6: Some studies have extended genetic-interaction mapping to different stress conditions (Martin et al. 2015; Gutin et al. 2015; St Onge et al. 2007; Díaz-Mejía et al. 2018), but not on a genome-wide scale. Therefore, the extent to which genetic interactions depend on environmental conditions and the feasibility of using additional phenotypes beyond colony growth phenotypes in genetic interaction mapping remains largely unexplored.

Page 22-23: Our genome-wide genetic interaction mapping in different conditions and cellular states indicates that: 1) genetic interactions with regulatory kinases vary between conditions; 2) genome-wide genetic interaction mapping is extensible to additional phenotypes and analyzing condition-specific phenotypes may increase the sensitivity for identifying novel regulatory relationships; and 3) for a given physiological state, increasing the number of conditions results in an increase in the number of significant genetic interactions. The first two points are consistent with studies in other organisms (Billmann et al. 2016, 2018; Fischer et al. 2015). This latter point is consistent with the limited number of studies that have investigated genetic interactions in different growth and stress conditions (Jaffe et al. 2019; Gutin et al. 2015; Martin et al. 2015). Despite the fact that our genetic interaction data set is limited in its scale and is focused on regulatory kinase genes, we anticipate that our methodology can be broadly applied to define genetic interactions in different conditions and cellular states.

Minor concerns:

1. The different influence of nutrient limitation and genetic background should be analyzed more directly (with regard to Figure 1B). Generalizing this concept will be important for the field but this manuscript currently does not seem to test for that specifically.

In this revision, we have undertaken additional comparisons of the correlation of fitness among the four mutant libraries in three nutrient starvation and two different cellular states (Fig 1E). We have also added a new paragraph expanding on this in results section (page 10):

The fitness of a genotype during proliferative growth in different media may differ from the survival of the genotype in response to a specific starvation signal. To test this, we separately modeled the relative abundance of each genotype during the growth phase (i.e. from $t = 0$ to $t = 24$ hours) and during the starvation period (i.e. from $t = 32$ to $t = 368$ hours) for all mutant libraries using all replicates (Table EV4). This analysis distinguishes the effect of each gene deletion in two distinct physiological states: proliferation and quiescence. As cells do not generate progeny when starved we refer to the phenotype during the starvation phase as “survival” and phenotype during proliferation as “fitness” (Fig 1D). To identify the primary determinant of these two phenotypes we quantified the similarity between fitness and survival for each mutant library in each condition (C, N, P restricted conditions) (Fig 1E & Table EV5). We find a clear distinction between proliferative and quiescent cells. During proliferation, mutant libraries starved for different nutrients tend to share similar fitness profiles regardless of the nutrient signals. By contrast, in quiescent cells, different mutant libraries starved for the same nutrient are more prone to have similar survival profiles. Consistent with the fitness estimation over the entire growth cycle, libraries starved for carbon have negative correlation with the other libraries starved for nitrogen and phosphorus.

2. The opposite fitness profile of carbon-restricted screens (with regard to Figure 1B) as compared to the other two media conditions is not immediately clear when looking at the heatmap in Figure 1B. The authors could visually support this statement by providing direct comparisons (e.g. scatter plots).

We now represent all pairwise comparisons of fitness profiles in FigEV2. We have also edited the results section to make this point clearer (page 9).

In general, mutants in carbon-restricted media show less similarity to that observed in nitrogen and phosphorus-restricted conditions, particularly for HO and PHO85 library (Fig EV2).

3. The authors mention both 3,600 genes and 4,800 strains on the array used. Please clarify how those numbers relate.

In our library construction, 4,800 single gene knock-out strains were used. However, after filtering out strains with low counts from our Bar-seq analysis, we studied ~4,000 that had sufficient counts. We have clarified this in the abstract, discussion and the results section (page 11):

The fitness of the single gene deletion mutants (methods) is distributed around 0 in each of the three proliferative conditions (Fig 2A), and the majority of mutants do not show a significant fitness defects compared to wild-type cells during proliferation (Fig 2A & Fig EV3B).

4. In the legend for Figure 1, descriptions for panels B and C are mixed up.

Thank you for identifying this error. We have corrected the panel descriptions.

Reviewer #2:

Summary and General Remarks:

The authors perform a genome-wide genetic interaction assay with three genes mediating quiescence. By using an informative phenotype for these genes, they find an enrichment of genetic interactions, and are better able to study the biology of cell quiescence. I thought that the two most interesting findings were that: 1) many genes that are dispensable for proliferative growth are required for quiescence, and 2) there is a lack of common quiescence-specific genes. Overall, I think this is a well-executed study and that the findings will be interesting to the readership of *Molecular Systems Biology*, and I recommend only minor revisions.

We thank the reviewer for the favorable assessment of our paper and for their constructive comments.

Major points:

My most pressing comment is on their finding that quiescence results in a five-fold enrichment of genetic interactions. It's previously been shown that genetic interactions are more common within a set of genes which are related to the phenotype being studied - e.g. St Onge et al 2007 find that when making double-knockout pairs of 26 genes that exhibit single-deletion sensitivity in MMS, 35% of gene pairs also have a genetic interaction in that condition. So, this enrichment may not be unprecedented. However, the authors also state that single-mutant effects predict genetic interaction propensity, and that quiescence results in increased variance in the distributions of survival compared to the distributions of fitness. While a greater interaction propensity amongst genes with greater single-mutant effects is likely to be at least partly biologically meaningful, it may also suggest that it is more difficult to model expected fitness when combining two single mutants with sizeable fitness defects. Thus, I am concerned that some GI enrichment may result from general deviations from either the GIS or the ANCOVA model (which are very similar) when combining large-effect mutations. Perhaps it may be possible for the authors to show that combining larger single-gene effects doesn't generally result in more deviations. If it does, they may control for the distribution of single-gene effects to see if an enrichment of interactions would still be present, or use a more empirical model for the combined effects of multiple mutations.

The increase in number of interactions in quiescent cells is striking. It is indeed true that genes exhibiting a fitness defect in a particular condition are more likely to show interactions with one another and with other genes in that condition as was

shown by St. Onge et al., and as is also the case in the large SGA datasets from Costanzo et al., Thus, the observation reported in our study is consistent with those previous studies.

However, we do not agree with the concern that the increased number of interactions is the result of noise as noise generally does not result in functional enrichment. Costanzo et al., 2016 showed that genetic interactions of genes with lower fitness are as functionally informative as the genetic interactions of genes with higher fitness. In our study, Figure 5 clearly shows that the different screens have strong functional enrichment profiles in quiescent cells and this would not occur if they were the result of increased random effects.

Minor points:

-Figure 1C (right panel) could use a more informative x-axis title than 'Coefficient of \ln '

We have changed this axis name to “Fitness (Coefficient)” to make it consistent with the legend on the heatmap.

-Captions for Figure 1B and C are swapped

Thank you for identifying this error. We have corrected the panel descriptions.

-Figure 2A is hard to read, I suggest the shading be removed

We have removed the gradient shading and changed the shading color.

-Figure EV2B: why do many more mutants result in better fitness in proliferation than result in defects? (43-50% vs 8-12%). This does not seem consistent with the violin plot in Figure 2B

We thank the reviewer for pointing this out. In addressing this point we discovered that an incorrect p-value had been applied to identify significant mutants. We have revised this analysis and generated a new Figure EV3B. The overall conclusion remains the same: namely, we detect an increase in the variance of the distributions in survival in comparison with the distributions of fitness in proliferation.

Reviewer #3:

Sun et al present a series of well designed experiments to investigate the effects of gene deletions on growth and quiescence under different nutrient limitations, and the role of epistasis in these biological processes.

The paper is scholarly and well-written, and I enjoyed reading it, and there are some interesting results in there. I do have a number of questions, largely to get clarification on some things I didn't quite understand, and there are also some issues that I enumerate below.

We thank the reviewer for the positive assessment of our manuscript.

Important things:

1. I was pretty shocked that some of the experiments did not cluster by medium - namely the TOR1 library in nitrogen, and the RIM15 in phosphorous. What makes it more surprising is that the TOR1 library in nitrogen looks pretty similar to the other libraries in Phosphorous, and RIM15 in phosphorous looks like the other libraries in carbon. I can wrap my head around that maybe they would look different, but having them cluster with libraries from another condition does raise the specter of there being some samples either misidentified/mixed up, or something funky happened to the media. Is there any way of post-facto determining that couldn't be the case?

This is indeed a surprising observation. However, we have been back through our raw data and analysis and can confirm that this is not the results of sample misidentification. We do note that the fitness profile of these two libraries - *PHO85* in nitrogen and *RIM15* in phosphorus - form distinct clusters compared to the other mutant libraries in the same condition. Our interpretation is that in these two cases the genetic background has a greater effect.

2. It struck me as odd to have the data clustered in Fig 1B, but for the paper to then say that it was much more useful (and I agree) to separate the time course into two pieces, to look at fitness during growth, vs. fitness during stationary phase, but that clustering of these data was not done - e.g. each deletion would have two values where it previously had one. I think clustering those data would be more useful than the combined data. Also, the contrast in Fig 1 was such that it's hard to see stuff. I think changing the contrast would help.

Our goal with Figure 1B is to show how nutrient and genotypes play a role in determining the overall fitness profiles that are comprised of both growth and quiescence. In addition, this global view visually depicts the quality of our data as all biological replicates from the same condition cluster together. Based on the reviewer's suggestions, we have changed the color scale for Fig 1B and performed additional comparison of the correlation of fitness among four mutant libraries in the three conditions and two different cellular states (Fig 1E) and our explanation (see new text added to reviewer #1 minor comment #1 above).

3. With regards to the experimental design, I do have concerns about the effects of drift and counting noise in assaying the abundance of the strains as they go through stationary phase. The paper says that they removed $2e5$ cells for expansion to count relative abundance. Given the number of strains in the pool, this is only ~ 40 /barcode, if they start out evenly, which is somewhat low. However, I think this gets exacerbated as time goes on, because viability drops during starvation - did they assess viability to know that the number of viable cells being plated was sufficient to avoid drift and counting noise?

The effect of sampling is one that we had not completely addressed in our original submission and we thank the reviewer for pointing this out. First, we corrected the number of cells used for outgrowth - in each case 5mL of a culture at $2e5$ cells/mL was used corresponding to $1e6$ cells. This corresponds to ~ 200 cells/barcode assuming they are present in equal abundance. As this is clearly not the case, we

computed the probability of losing a genotype due to sampling as a function of different mutant abundances. The average sequencing library size in our sequencing run is $\sim 1e6$ reads, and barcode counts lower than 4 were removed (see methods). Therefore, the lowest genotype frequency that we consider in our assay is $> 4e-6$, and the chance of losing this genotype at this frequency is 1.8%. We now include this analysis in Fig EV1D.

In addition, we did track population viability, which shows no substantial change in each population and have added these data as panel E in Figure EV1. We refer to these results on page 8-9:

To compare the fitness of each genotype over the complete growth cycle in each condition, a 1mL sample (1×10^6 cells) was removed from the culture at sequential time points and the subpopulation of viable cells was expanded using 24-48 hours of outgrowth in supplemented minimal media (Fig 1A, methods). This step is required to enrich for mutants that survive proliferation and starvation and to deplete those that have undergone senescence. Sampling 1×10^6 cells from the cultures minimized the probability ($P < 0.018$) that a genotype was not measured due to sampling error (Fig EV1D and methods). We also quantified population viability throughout this period and observed to no substantial change in any of the conditions (Fig EV1E).

4. I'm presuming that the entire section "Distinct cellular functions...", and the data in Figure 2, are based on just the control data in the HO background, but it should be made explicit (if it's not the case, I would like to know why).

We have altered the text to make this more explicit (page 11).

To test whether the genetic requirements for proliferation in nutrient-restricted media and quiescence in response to starvation for the same nutrient are distinct, we investigated the fitness and survival of each genotype in the single mutant library (i.e. the *HO* library).

5. The authors say that many of the genes important for quiescence show no fitness effects in proliferation, but I think I disagree. RGT2 and GPR1 seem to show profound fitness increases during the proliferative phase, suggestive of a trade-off. The same is true (if not more so) for the genes involved in phosphorus limitation, where those pointed out appear to be the most fit, including two other RIM genes, so maybe I misunderstood something.

We have edited the results section (page 11) to include this observation.

Critically, many of the genes that are dispensable for proliferative growth in each of the three media conditions are required for quiescence. For example, deletion of genes involved in the cAMP-PKA signaling pathway, GPB1/2, RGT2, GPR1 results in a profound survival defect in response to carbon starvation, but deletion of these genes does not lower the fitness of carbon-restricted proliferating cells and actually appear to result in a fitness increase, suggestive of a trade-off (Fig 2A left-panel).

6. I didn't really understand why the authors chose as the interesting genes in quiescence those that have a difference between their quiescent and proliferative fitness (Squi - Fpro) - why isn't just survival (or lack thereof) the important quantity? It seems like this will change the analysis substantially, and thus the conclusions, so I want more of a rationale as to why it isn't simply sufficient to ask which survive the worst during stationary phase.

Our rationale is that our goal is to identify the genes that are specifically required for maintaining viability in quiescence. Therefore, our approach is to filter out those genotypes who have already shown a defective fitness in proliferation by taking the difference between the two phenotypes.

7. For determining Squi, did the authors use a different starting timepoint for the carbon limitation, given that growth persists for longer? It seems like that should be the case.

We had tried to use a later initial timepoint (48hrs) rather than 32hrs for quiescence, but found that the survival rate for each mutant is not different. Therefore, to simplify the data analysis, we used 32hrs as the initial timepoints for all starvations.

Minor issues:

1. Appendix_Fig1 does not exist - I looked through the uploaded files, but couldn't find it

We apologize. We have now moved this figure to FigEV1F and edited the text in the results section on page 9.

To test the reproducibility of our fitness assay, we first estimated fitness for each biological replicate separately and used PCA analysis to identify and exclude poorly behaved libraries (Fig EV1F).

2. The graph in Figure 1 has an x-axis labeled as "hours post nutrient restriction", but I would prefer as "hours post inoculation"

We have changed the original label to "hours post inoculation".

3. I think it's pretty well known that genes involved in the Ras pathway improve fitness during growth under carbon limitation, but that they exhibit severe trade-offs in stationary phase, so earlier work to that effect should be cited (I can find references if the authors are unable to do so)

We have added references to the role of loss of Ras signaling in enhancing fitness under carbon limitation. (page 11)

Critically, many of the genes that are dispensable for proliferative growth in each of the three media conditions are required for quiescence. For example, deletion of genes involved in the cAMP-PKA signaling pathway, GPB1/2, RGT2, GPR1 results in a profound survival defect in response to carbon starvation, but deletion of these genes does not lower the fitness of carbon-

restricted proliferating cells and actually appear to result in a fitness increase, suggestive of a trade-off (Fig 2A left-panel). This observation is consistent with the fact that mutations in cAMP-PKA pathway have increased fitness in carbon-limiting conditions (Venkataram et al. 2016).

4. Figure 5 is pretty hard to read, especially the key. Please play with the color scheme to come up with something more interpretable.

We have revised Figure 5 with less conditions in the main figure with increased the font size in the key. The figure for other conditions have been removed to Fig EV8B.

5. The data in EV4A are pretty cool/interesting, but I'd also like to see the actual clustered data (maybe with some zoom ins to interesting groups of genes).

We have added zoom-in scatter plots as a new supplementary figure (Fig EV7) for representing the interesting and representative gene clusters. We have also added the interpretation of the results (page 17) as follows:

For example, a negative correlation is detected for *TOR1* and *PHO85* in proliferative cells growing in carbon-restricted condition, but their genetic interaction profiles are positively correlated in carbon-starved quiescent cells (Fig 4B & Fig EV7B). For cells in the same physiological state, the environmental conditions can also alter the functional relationship between the same pair of kinases. For example, *RIM15* and *PHO85* genetic interaction profiles are highly correlated during growth in carbon-restricted media, but this similarity is greatly reduced during proliferation in phosphorus-restricted conditions (Fig 4A & Fig EV7C). These results suggest that environmental conditions alter the regulatory relationships among signaling pathways both in quiescent and proliferative cells.

Trivial points:

In the Introduction, "ethanol as carbon source" should be "ethanol as the carbon source"

We have made this change.

In the Results, "mutants was used to inoculate" should be "mutants were used to inoculate", as were refers to "cells"

We made this correction.

In Figure 1, the legends for B and C are switched (or their labeling on the Figure is switched)

We thank the reviewer for identifying this error. We have corrected the panel descriptions.

Gene names in the Methods should be italicized

We have used italics when referring to the gene and roman when referring to the protein throughout the text.

2nd Editorial Decision

17th February 2020

Thank you again for sending us your revised manuscript. We have now heard back from the reviewer who was asked to evaluate your study. As you will see below, the reviewer acknowledges that the study has improved as a result of the performed revisions. They raise however a few remaining concerns, which we would ask you to address in a minor revision.

REFEREE REPORTS

Reviewer #1:

In the initial review, we stated the potential but also several shortcomings of this study. We focused our critique on aspects of data analysis. The authors have addressed most concerns well. However, we would like to specify 3 major concerns and re-enforce their importance for this work:

Major concerns (referring to the point-by-point response of reviewer #1):

1. The absence of correlation of fitness in proliferation and survival is an important side-note and the expectation is somewhat open. Specifically, positive, negative or absence of correlation might be equally biologically relevant and insightful. We appreciate that the authors clarified this relationship but believe that a more specific description ("poorly correlated" is not quantitatively relevant) could add clarity.
2. This analysis addresses our concern.
3. This edit addresses our concern.
4. This analysis partly addresses our concern. Please also see major concern 5 for the importance of a reasonable model.
5. We appreciate the author's response and agree with the challenges when interpreting (significant) genetic interactions. However, identifying many significant events with low enrichment and statistically weak signal with high enrichment might point to substantial issues in the data (due to experimental and/or analytical issues). This could be highly relevant for this study as it confounds a large portion of the interpretations.
6. We would like to re-emphasize that this concern addressed the lack of context in this studies narrative with regard to previous studies of CONTEXT-specific (where context refers to a specific measured phenotype or environmental condition) genetic interaction studies. The fact that other (non-yeast) genetic models have been explored is not important here.

Minor concerns:

All minor concerned addressed.

2nd Revision - authors' response

18th March 2020

Please find below our response to each point raised by the reviewer. Our responses are indicated in **red** and major changes made to the text are excerpted here and indicated in **bold**.

Reviewer #1:

In the initial review, we stated the potential but also several shortcomings of this study. We focused our critique on aspects of data analysis. The authors have

addressed most concerns well. However, we would like to specify 3 major concerns and re-enforce their importance for this work:

We thank the reviewer for recognizing our efforts to address the comments we received on our manuscript. We are grateful for their careful reading of our revised manuscript and their additional comments. We have addressed these important comments as detailed below, which has helped us to further clarify our findings.

Major concerns (referring to the point-by-point response of reviewer #1):

1. The absence of correlation of fitness in proliferation and survival is an important side-note and the expectation is somewhat open. Specifically, positive, negative or absence of correlation might be equally biologically relevant and insightful. We appreciate that the authors clarified this relationship but believe that a more specific description ("poorly correlated" is not quantitatively relevant) could add clarity.

We have clarified this finding by reporting the numerical values of the low correlation coefficients in the main text as follows:

Page 10: “We find that fitness in proliferation and survival in quiescence are poorly correlated for all three nutrient-restricted media: pearson $r = -0.033$ in carbon restricted condition, 0.052 in nitrogen restricted conditions, and 0.064 in phosphorus restricted conditions (Fig EV4A).”

In addition, we provide a visual depiction of the quantitative correlation between all pairs of conditions (Fig 1E). The underlying raw data are depicted as a matrix of pairwise scatter plots (Fig EV3) with the following text:

Page 9: “To identify the primary determinant of these two phenotypes we quantified the similarity between fitness and survival for each mutant library in each condition (C, N, P restricted conditions) (Fig 1E). We find a clear distinction between proliferative and quiescent cells. During proliferation, mutant libraries starved for different nutrients tend to share similar fitness profiles regardless of the nutrient signals (Fig 1E - lower left). By contrast, in quiescent cells, different mutant libraries starved for the same nutrient are more prone to have similar survival profiles than those starved for different nutrients (Fig 1E - upper right). Consistent with the fitness estimation over the entire growth cycle, libraries starved for carbon have negative correlation with the other libraries starved for nitrogen and phosphorus (Fig 1E & Fig EV2)”

Finally, to further understand the functional specificity in proliferating and quiescent cells, we have applied GSEA to the fitness profiles of single mutant libraries in both cellular states (Fig EV4C) with the following interpretation:

Page 11: “To further investigate the functional relationship between proliferating and quiescent cells, we applied GSEA to the fitness and survival profiles of the single mutant library in each nutrient-restricted condition. We find no functional overlap between different cellular states under the same nutrient-restricted condition (Fig EV4C). Moreover, in many cases the same set of genes has the opposite behavior in fitness and survival. For example,

deletion of genes involved in protein deacetylase activity shows no significant impact on survival in quiescent cells, but results in reduced fitness during proliferation in nitrogen-restricted conditions (Fig EV4C)."

2. This analysis addresses our concern.

Thank you.

3. This edit addresses our concern.

Thank you.

4. This analysis partly addresses our concern. Please also see major concern 5 for the importance of a reasonable model.

Please see below.

5. We appreciate the author's response and agree with the challenges when interpreting (significant) genetic interactions. However, identifying many significant events with low enrichment and statistically weak signals with high enrichment might point to substantial issues in the data (due to experimental and/or analytical issues). This could be highly relevant for this study as it confounds a large portion of the interpretations.

We believe that this comment reflects our failure to adequately describe our different methods for defining enrichment, which we have now clarified. We have performed functional enrichment analysis of significant genetic interactions *only*. We find functional enrichment for genetic interactions in quiescent conditions, which have many more significant interactions than proliferative conditions (Fig 6 and Fig EV12) for which we find no significant enrichment. We have revised the manuscript to read:

Page 18: "Using SAFE analysis, we find that *RIM15* consistently interacts with genes functioning in multivesicular bodies (MVB) sorting and pH-dependent signaling under all starvation conditions in both cell types (Fig 5). This suggests that *RIM15* plays an essential role in regulating protein homeostasis via MVB sorting. As the reference genetic interaction map used for SAFE does not include all genes in our genetic interaction dataset (only ~2,900 non-essential genes are present in the reference) and tests only for coherence of both statistically significant and non-significant interactions, we performed over-representation analysis on the sets of genes that significantly interact with each kinase (method and materials). Due to the limited number of significant interactions detected in proliferative cells (Fig 3D and Fig EV5D), we did not find any enriched GO terms for kinases in proliferative cells. However, we identified multiple significantly enriched functional categories in quiescent cells."

Second, we performed a SAFE analysis. This method of analysis quantifies the *coherence* of genetic interactions within functional domains of a reference map defined by the genetic interaction similarity network from the Costanzo et al., 2016.

Specifically, SAFE assesses whether genetic interactions tend to be positive or negative within a “domain” regardless of significance. Using this analysis, we do find that domains are enriched for genetic interactions in both quiescent and proliferative conditions despite the relative paucity of significant genetic interactions in proliferative conditions. It is important to note that this reference map is 1) defined by genetic interactions quantified using fitness in rich media conditions and 2) the domains in this map are only enriched for 12% of all possible GO terms (see Baryshnikova, Cell Systems, 2016), suggesting that many functions are not captured in the network. Therefore, the presence or absence of enrichment within this map reflects both the distribution of genetic interactions within each dataset and the overlap in functional importance with the reference map. To interpret this result we have added each of the raw plots for SAFE analysis (Fig EV12) and added the following text:

Page 17: “Importantly, SAFE uses the entire set of genetic interactions for a given query gene, including those interactions that do not reach statistical significance, which allows identification of coherent trends that may exist despite a lack of significance associated with each individual genetic interaction. This analysis tests specifically for coherence in attributes such that strong positive and negative genetic interaction scores that are randomly distributed throughout the network will produce no enrichment, whereas weak scores that tend to cluster as either positive or negative scores within domains will have significant enrichment. We superimposed genetic interaction profiles of each kinase in each of the three nutrient-restricted media and both cellular states onto the reference network using SAFE. We find that kinases that show higher similarity in genetic interaction profiles (Fig 4) also show more similar enrichment patterns using SAFE analysis (Fig 5). In general, genetic interactions in proliferative conditions tend to show increased enrichment when superimposed on this reference map indicating greater similarity among positive or negative interactions within each domain despite the relative paucity of significant interactions (Fig EV12). This difference may reflect the fact that genetic interactions in quiescent cells reflect novel regulatory relationships compared with those identified using fitness measurements in rich media that were used to construct the reference map.”

Finally, to address the reviewer’s concern about the data quality, we have performed multiple additional analyses. First, we have ruled out the possibility of systematic noise in using ANCOVA for GIS quantification, because the results (GI profiles and GSEA) generated using different statistical models (classical multiplicative model and ANCOVA model) are highly correlated. Second, we have added diagnostic plots to show that there is no systematic bias in our analytical method (Fig EV6). Third, our results, generated using ANCOVA, are consistent with Costanzo et al., 2016 study in that single mutants with stronger defects tend to have positive interactions (Fig EV7). We have added the following text describing these additional analyses:

Page 13-14: “To further compare the two approaches, we applied GSEA to genetic interaction profiles calculated using each model and compared the similarity of generated GO terms using GoSemSim (Yu et al. 2010). The significant GO terms for a given condition identified using the different models

are very similar (Table EV3), indicating that ANCOVA identifies the same genetic interactions and functional groups as the classic multiplicative model. An analysis of estimated effect sizes and standard error indicates that there are no systematic biases in applying the ANCOVA model (Fig EV6). As ANCOVA has a well developed statistical framework for error estimation and significance testing, we elected to use ANCOVA to compute GIS for all subsequent analyses. As has been previously observed (Costanzo et al. 2016), genes with larger phenotypic effects (either fitness or survival) tend to have strong interactions (Fig EV7).”

Page 18: “We also found cases of functional enrichment that are maintained in the two different cellular states. For example, genes involved in peroxisome functions are enriched for negative interactions with *PHO85* in carbon restricted proliferative cells and carbon starved quiescent cells (Fig 5, cyan arrow/circle). This is consistent with the known role of *PHO85* in regulating long-chain based kinase during stationary phase (Iwaki et al. 2005) suggesting that *PHO85* may play a role in maintaining long-chain fatty acid recycling and provide energy for cells in calorie-restricted conditions. ”

6. We would like to re-emphasize that this concern addressed the lack of context in this studies narrative with regard to previous studies of CONTEXT-specific (where context refers to a specific measured phenotype or environmental condition) genetic interaction studies. The fact that other (non-yeast) genetic models have been explored is not important here.

We have revised the text to emphasize previous studies that have investigated context specific genetic interactions and described the contribution of studies in other non-yeast genetic models.

Page 21: “To date, genome-wide genetic interaction mapping in yeast has primarily been assayed using a single phenotype in a single condition - colony growth in rich media. Our genome-wide genetic interaction mapping in different conditions and cellular states indicates that: 1) genetic interactions with regulatory kinases vary between conditions; 2) genome-wide genetic interaction mapping is extensible to additional phenotypes and analyzing condition-specific phenotypes may increase the sensitivity for identifying novel regulatory relationships; 3) less favorable conditions result in an increased number of significant interactions; and 4) for a given physiological state (e.g. proliferation or quiescence), increasing the number of environmental conditions results in an increase in the number of significant genetic interactions. The points are consistent with, and extend, the limited number of studies that have investigated genetic interactions in different growth and stress conditions (St Onge et al. 2007; Gutin et al. 2015; Martin et al. 2015). Despite the fact that our genetic interaction data set is limited in its scale and is focused on regulatory kinase genes, we anticipate that our methodology can be broadly applied to define genetic interactions in different conditions and cellular states.”

Accepted

31st March 2020

Thank you again for sending us your revised manuscript. We are now satisfied with the modifications made and I am pleased to inform you that your paper has been accepted for publication.

Corresponding Author Name: David Gresham

Journal Submitted to: Molecular and System Biology

Manuscript Number: MSB-19-9167